# The GNN Trilemma in Recommender Systems: A Survey

## Abstract

Graph Neural Networks (GNNs) have become the standard choice for modeling collaborative interactions in recommender systems via message passing. However, as industrial deployments scale, traditional static GNNs face fundamental limitations, including noise propagation, semantic rigidity, and computational bottlenecks. Recent advances (2024–2026) reveal a convergence of generative refinement (e.g., diffusion models) and semantic hybridization (e.g., Large Language Models) to address these challenges. In this survey, we systematically analyze this architectural shift. We introduce an orthogonal three-axis taxonomy that categorizes models along Information Source, Learning Paradigm, and System Objectives. In doing so, we capture the transition from heuristic structural augmentations toward dynamic, privacy-aware, and objective-aligned frameworks. To analyze these design trade-offs, we introduce the *GNN Trilemma*, a structural framework that examines how improvements in accuracy, scalability, and explainability often compete with one another in practical recommender architectures. Finally, we argue that a growing evaluation crisis exists: as models optimize for complex human-centric objectives, traditional static benchmarks and simplistic heuristic baselines increasingly obscure true system-level trade-offs.

## 1 Introduction

The integration of Graph Neural Networks (GNNs) into recommender systems shifted how collaborative filtering is approached. By representing users and items as nodes and historical interactions as edges, models such as LightGCN successfully captured high-order collaborative signals through neighborhood aggregation (He et al., 2020). However, many standard message-passing approaches rely on relatively static interaction graphs and limited semantic signals. In real-world environments, these graphs suffer from popularity bias, noisy interactions, and lack the rich contextual meaning behind user behaviors.

Between 2024 and 2026, researchers have actively explored alternatives to traditional message passing. We observe a trend toward Generative Refinement, where continuous-time differential equations and asymmetric diffusion models denoise sparse interaction graphs. Researchers also explore Semantic Hybridization, leveraging Large Language Models (LLMs) to inject semantic reasoning into structurally constrained graph representations.

### 1.1 The Gap in Existing Surveys

As shown in Table 1, existing survey literature remains fragmented across several important research directions. Broad surveys of GNN-based recommendation primarily focus on architectural taxonomies and foundational graph learning techniques (Zhang et al., 2023; Anand & Maurya, 2024), but were published before the rapid emergence of diffusion-based recommendation and LLM-integrated frameworks. In contrast, recent surveys dedicated to LLM-based recommendation and graph foundation models provide detailed discussions of semantic reasoning and large-scale pretraining (Shehmir & Kashef, 2025; Wu et al., 2025a), yet give comparatively limited attention to deployment-oriented challenges such as machine unlearning, privacy preservation, and evaluation under real-world system constraints. Industrial recommender-system surveys provide valuable insights into scalability and production deployment (Zou & Sun, 2025), but they do not primarily examine how these challenges influence the design of modern GNN architectures.

Table 1: Comparison of recent recommender-system surveys with our survey.

| Survey | Emerging Learning Paradigms | | User Dynamics | System Design | Trustworthiness | Responsible AI | Evaluation |
|---|---|---|---|---|---|---|---|
| | LLM-Enhanced Recommendation | Diffusion / Generative Models | Temporal and Dynamic Modeling | Scalable Architectures | Privacy and Machine Unlearning | Fairness and Explainability | Evaluation Critique |
| Recommending on Graphs: A Comprehensive Review (2023) (Zhang et al., 2023) | ✗ | ✗ | ● | ● | ✗ | ● | ● |
| A Survey on Recommender Systems using Graph Neural Networks (2024) (Anand & Maurya, 2024) | ✗ | ✗ | ● | ● | ✗ | ● | ✗ |
| Review of Explainable Graph-Based Recommender Systems (2025) (Markchom et al., 2025) | ✗ | ✗ | ✗ | ✗ | ✗ | ✓ | ✓ |
| A Survey of Real-World Recommender Systems (2025) (Zou & Sun, 2025) | ● | ✗ | ● | ✓ | ● | ● | ✓ |
| LLM4Rec: A Comprehensive Survey on LLM Integration (2025) (Shehmir & Kashef, 2025) | ✓ | ✓ | ● | ● | ✗ | ● | ● |
| Graph Foundation Models for Recommendation (2025) (Wu et al., 2025a) | ✓ | ● | ● | ● | ✗ | ● | ● |
| **Our Survey** | ✓ | ✓ | ✓ | ✓ | ✓ | ✓ | ✓ |

**Legend:** ✓ Fully covered, ● partially discussed, ✗ not explicitly addressed.

This survey integrates these perspectives within a unified framework. We analyze recent advances in GNN-based recommendation across emerging learning paradigms, temporal modeling, scalable architecture design, security and privacy, objective-aligned learning, and evaluation practices. By connecting these developments through a common systems-oriented perspective, we provide a comprehensive view of both recent progress and the challenges that continue to influence real-world deployment.

## 1.2 The GNN Trilemma

To provide a rigorous critique of modern architectures, we introduce the *GNN Trilemma*. Analogous to the trade-offs formalized by the CAP theorem in distributed systems, modern GNN-based recommenders exhibit an inherent structural and mathematical tension between three system-level properties. We operationalize these properties through their architectural mechanisms and theoretical constraints:

**Accuracy:** The structural capacity to capture deep collaborative signals, multi-hop semantic relationships, and continuous temporal dynamics. While traditionally validated via empirical ranking metrics (e.g., NDCG@K and Recall@K), it fundamentally relies on the mathematical expressiveness and hypothesis space of the underlying architecture.

**Scalability:** The ability to execute inference and training under strict industrial hardware constraints. To ensure hardware-agnostic evaluation, scalability in this survey is primarily analyzed through theoretical computational complexity (e.g., bounding message passing at $\mathcal{O}(|E|d)$ versus absorbing quadratic $\mathcal{O}(|V|^2)$ bottlenecks) and structural memory footprints, rather than environment-dependent latency metrics.

**Explainability:** The architectural ability to provide interpretable or semantically meaningful rationales for recommendations. This property is evaluated by the presence of structural transparency, ranging from discrete behavior tokenization and explicit ego-path extraction to interpretable attention mechanisms and LLM-generated priors.

While novel architectures frequently push the boundaries of one or two of these dimensions, they mathematically necessitate compromises in the third. For instance, recent work explores alternatives to localized message passing using global attention mechanisms (Ying et al., 2021); while this substantially improves the modeling of long-range dependencies (Accuracy), it inevitably introduces severe scalability bottlenecks due to the quadratic complexity of global self-attention.

## 1.3 Contributions

This survey offers the following core contributions:

- We propose a novel, orthogonal 3-axis taxonomy (Information Source, Learning Paradigm, System Objectives) to categorize over 30 recent architectures published between 2024 and 2026.

- We introduce and operationalize the *GNN Trilemma* to systematically evaluate the structural, theoretical, and algorithmic trade-offs inherent across modern graph-based recommender architectures.

- We expose a critical evaluation crisis in the field, demonstrating how the continued reliance on static offline datasets and traditional ranking metrics fails to capture the true deployment costs, latency constraints, and dynamic adaptation requirements of modern systems.

## 2 Survey Methodology

To ensure a focused and technically grounded review, we conducted a structured analysis of recent research in GNN-based recommender systems. The survey emphasizes methodological developments related to semantic modeling, generative learning, scalability, and objective-aware recommendation.

### 2.1 Literature Scope

The primary focus of this survey is research published between 2024 and 2026. We collected papers from major academic repositories and indexing platforms, including the ACM Digital Library, IEEE Xplore, OpenReview, and arXiv, with particular attention to venues such as SIGIR, KDD, WSDM, WWW, AAAI, NeurIPS, and RecSys.

Given the rapid publication cycle of modern recommender systems research, especially in diffusion-based and LLM-enhanced methods, we additionally included a limited number of recent arXiv preprints. Preprints were considered only when they provided sufficient methodological detail, comprehensive empirical evaluation, and publicly available reproducibility artifacts.

### 2.2 Paper Selection Criteria

Candidate papers were selected based on their relevance to GNN-based recommendation and their contribution beyond incremental architectural modification. We prioritized works that introduced at least one of the following:

- novel graph representation or semantic modeling strategies,

- new learning paradigms such as diffusion, contrastive, continual, or continuous-time learning,

- scalable or efficiency-oriented architectural designs,

- privacy, fairness, explainability, or objective-aware optimization mechanisms,

- integration of semantic reasoning components such as LLMs or knowledge graphs.

We excluded purely non-graph recommendation models, papers lacking sufficient methodological clarity, and works that introduced only minor layer-level modifications without broader implications for scalability, representation quality, or system behavior.

### 2.3 Survey Organization and Taxonomy Development

The taxonomy was developed iteratively through comparative analysis of recent GNN-based recommender architectures and their learning objectives. Rather than grouping methods solely by model family, we organize the literature along three complementary axes: information and semantic representation, learning paradigm, and system-level objectives.

This organization is intended to capture the broader methodological shifts in modern recommender systems research, including semantic augmentation, generative modeling, scalable lightweight architectures, and objective-aware optimization. In addition to recent architectures, we reference a smaller set of foundational pre-2024 works to provide historical context for graph message passing, collaborative filtering objectives, and commonly used evaluation protocols.

# 3 Background and Preliminaries

To contextualize the architectural trade-offs discussed in the remainder of this survey, we first formalize the standard graph-based recommendation paradigm and establish the primary evaluation metrics used across the literature.

## 3.1 Graph Recommendation Preliminaries

At its core, a collaborative filtering recommender system can be modeled as a bipartite graph $\mathcal{G} = (\mathcal{U} \cup \mathcal{I}, \mathcal{E})$, where $\mathcal{U} = \{u_1, u_2, \ldots, u_M\}$ represents the set of users, $\mathcal{I} = \{i_1, i_2, \ldots, i_N\}$ represents the set of items, and $\mathcal{E}$ represents the observed interactions (e.g., clicks, purchases) between users and items. While bipartite graphs captures direct interactions, many modern systems enhance this structure by incorporating external facts and details. A Knowledge Graph (KG) serves this purpose by mapping out rich semantic relationships. Formally, a KG is a directed graph composed of entity-relation-entity triplets, commonly denoted as $(h, r, t)$. In this notation, $h$ and $t$ represent the head and tail entities (such as a specific movie and its director), while $r$ defines the specific relation connecting them (such as "directed by"). By merging a KG with a standard user-item graph, models can leverage these semantic relationships to generate more accurate and explainable recommendations.

The goal of Graph Neural Networks in this setting is to learn high-quality $d$-dimensional vector representations (embeddings) for both users and items. Let $\mathbf{e}_u^{(0)} \in \mathbb{R}^d$ and $\mathbf{e}_i^{(0)} \in \mathbb{R}^d$ denote the initial ID embeddings for user $u$ and item $i$, respectively. Traditional architectures, such as Neural Graph Collaborative Filtering (NGCF) (Wang et al., 2019) and LightGCN (He et al., 2020), refine these representations through a multi-hop message-passing paradigm.

At each layer $k$, a node updates its representation by aggregating features from its local topological neighborhood. The simplified, linear aggregation defined by LightGCN—which serves as the baseline for most modern architectures—is formulated as:

$$\mathbf{e}_u^{(k+1)} = \sum_{i \in \mathcal{N}_u} \frac{1}{\sqrt{|\mathcal{N}_u||\mathcal{N}_i|}} \mathbf{e}_i^{(k)}$$

where $\mathcal{N}_u$ and $\mathcal{N}_i$ denote the neighbors of user $u$ and item $i$. After $K$ layers of propagation, the final representations are typically obtained by pooling the embeddings across all layers. The predicted preference score $\hat{y}_{ui}$ of user $u$ for an unobserved item $i$ is then calculated via the inner product:

$$\hat{y}_{ui} = {\mathbf{e}_u^{(K)}}^\top \mathbf{e}_i^{(K)}$$

## 3.2 Common Evaluation Metrics

Modern recommender systems are typically evaluated under a Top-$K$ ranking protocol. The model generates preference scores for all unobserved items, ranks them in descending order, and recommends the top $K$ items to the user. To quantify the quality of this ranked list, the community relies on several standardized metrics:

**Recall@$K$:** Recall measures the proportion of a user's actual positive interactions (ground truth) that successfully appear within the top $K$ recommendations. It evaluates the model's raw retrieval capability without penalizing the specific order of items within that top $K$ window.

**Precision@$K$:** Precision measures the proportion of recommended items within the top $K$ list that are actually relevant to the user. It is calculated by dividing the number of relevant items appearing in the top $K$ recommendations by $K$. Unlike Recall, Precision emphasizes recommendation accuracy by rewarding models that minimize irrelevant suggestions in the ranked list.

**Normalized Discounted Cumulative Gain (NDCG@$K$):** Unlike Recall, NDCG is highly position-aware. It assigns higher rewards to relevant items that appear at the very top of the ranking list. The metric applies a logarithmic discount factor to items further down the list, reflecting the reality that users rarely scroll past the first few recommendations.

**Hit Ratio (HR@$K$):** Hit Ratio is a binary, user-level metric. If at least one relevant item appears in the user's top $K$ recommendation list, the HR is 1; otherwise, it is 0. The global HR is the average across all users. It is particularly dominant in "leave-one-out" evaluation settings where the model is tested on predicting a single hidden target item.

**Mean Reciprocal Rank (MRR):** MRR focuses exclusively on the rank position of the *first* relevant item the model recommends. It is calculated as the average of the inverse ranks (1/rank) across all users. MRR strongly penalizes models that fail to place a highly relevant item at the absolute top of the list.

**Average Precision (AP) & Mean Average Precision (MAP):** Average Precision summarizes the precision-recall curve by calculating the precision at every position where a relevant item is found, and then averaging those values. MAP is simply the mean of the AP scores across all users. It is an extremely rigorous metric that requires the model to correctly rank all relevant items ahead of irrelevant ones.

While these offline ranking metrics form the foundation of academic benchmarking, industrial deployment introduces additional constraints—such as inference latency, memory overhead, and online Click-Through Rate (CTR), which we critically analyze in Section 6.

## 4 The GNN Trilemma: Architectural Pressures and Trade-offs

As established, modern GNN-based recommenders face a fundamental tension between Accuracy, Scalability, and Explainability. While traditional models like LightGCN (He et al., 2020) occupy a compromise in the center of this space, the architectures developed between 2024 and 2026 aggressively optimize for specific extremes.

By analyzing the theoretical limitations of these model families (summarized in Table 3) alongside their structural mechanisms and mathematical complexity constraints (detailed in Table 2), we can observe the physical and mathematical manifestations of the GNN Trilemma.

### 4.1 The Conflict Between Accuracy and Scalability

The most dominant trade-off in modern recommendation systems is the tension between predictive accuracy and deployment scalability. To achieve maximum accuracy, models must structurally capture deep topological signals or continuous temporal dynamics. However, doing so introduces mathematical complexities that inherently conflict with strict industrial latency constraints.

As detailed in Table 2, achieving high temporal expressiveness often requires abandoning lightweight discrete decoders. For instance, TGODE (Fu et al., 2025) decouples time and evolution graphs via Neural Ordinary Differential Equations (ODEs), but incurs the substantial computational cost of iterative Runge-Kutta numerical integration. Similarly, DiffNBR (Zhang et al., 2026b) utilizes Denoising Diffusion Probabilistic Models (DDPMs) to encode spatial and temporal evolution. While this generative denoising yields notable offline accuracy gains, it introduces severe quadratic self-attention bottlenecks relative to the sequence length ($|S|$) and basket size ($|B|$), resulting in a computational bound of $\mathcal{O}(|U|(|S|^2+|B|^2))$ that substantially limits its scalability for real-time inference.

Conversely, architectures designed for extreme scalability must structurally discard parts of the graph topology. Graph-free paradigms, such as AlphaFree (Jeon et al., 2026), achieve a 69% reduction in online GPU memory by abandoning $K$-hop message passing entirely, bounding inference to a highly efficient $\mathcal{O}(d_{LR}dn + n \log K)$. However, this requires absorbing a heavy quadratic $\mathcal{O}(n^2)$ offline preprocessing bottleneck. Similarly, PULSE (Choi et al., 2026) addresses parameter bottlenecks by entirely eliminating the standard $\mathcal{O}(|U|d)$ user embedding table, dynamically generating representations from community affiliations instead. While these lightweight architectures achieve exceptional inference speeds—such as the 50× speedup achieved by EAGLE (Li et al., 2025a) over traditional temporal GNNs—they fundamentally limit deep, localized multi-hop expressiveness.

### 4.2 The Conflict Between Explainability and Scalability

Providing human-readable explanations for recommendations is computationally antagonistic to highly parallelized matrix multiplications that make standard GNNs efficient.

To achieve high explainability, models must either extract discrete subgraph routing paths or generate semantic rationales using Large Language Models (LLMs). Both approaches severely degrade inference speed. For example, Table 2 shows that while SoREX (Guo et al., 2025) achieves high explainability by explicitly extracting structural motifs like "co-purchase" ego-paths, its reliance on real-time random walk sampling bounds its complexity at $\mathcal{O}(L|E|d + n_w|E_r|d)$ (where $n_w$ denotes the number of random walks). This structural requirement drops inference speed to one or two orders of magnitude slower than the LightGCN baseline.

Similarly, LLM-hybrid systems successfully fuse semantic reasoning with recommendation, but must execute complex architectural maneuvers to avoid prohibitive latency. RecMind (Wang et al., 2024) and BEAT (Feng et al., 2026) achieve deep semantic transparency, but strictly avoid end-to-end LLM fine-tuning. Instead, they shift the mathematical bottlenecks to offline preprocessing stages: BEAT relies on offline dual-codebook quantization to construct discrete behavior vocabularies, whereas RecMind utilizes offline textual encoding together with lightweight LoRA adapters. These design choices force the models to reason over compressed representations while maintaining practical system scalability.

### 4.3 The Conflict Between Accuracy and Explainability

A subtle but persistent tension exists between raw ranking accuracy and explainability. The most accurate collaborative filtering models rely on dense, unconstrained, continuous embedding spaces where features are highly entangled.

When researchers force a model to be explainable—such as constraining representations to a discrete behavior vocabulary as seen in BEAT (Feng et al., 2026), or mathematically restricting information flow to human-readable ego-paths like SoREX (Guo et al., 2025)—they inherently restrict the model's hypothesis space. Objective-aligned models like TopKGAT (Chen et al., 2026b) explicitly sacrifice generic graph-smoothing uniformity to focus strictly on borderline candidates via band-pass activation. While these structural constraints are necessary for transparency and targeted ranking, they often prevent the network from capturing the abstract, latent correlations that drive the highest empirical ranking scores in pure, unconstrained black-box models.

## 5 A Unified Taxonomy of Graph Recommendation Architectures

To systematically analyze the rapid evolution of GNN-based recommendation, we propose a unified 3-axis taxonomy, visualized in Figure 1. Figure 1 illustrates how these taxonomic advancements act as modular plug-ins mapped directly onto the standard recommender system pipeline. We categorize the recent literature based on their primary architectural innovations: the information source they ingest (Axis 1), the generative or continuous learning paradigms they employ (Axis 2), and the deployment objectives they optimize for (Axis 3).

*Disclaimer on Scope:* Modern recommender architectures are inherently multi-faceted. A single framework may leverage knowledge graphs (Axis 1) while simultaneously optimizing for fairness constraints (Axis 3). In this taxonomy, models are categorized based on their primary architectural contribution or most dominant optimization objective, though their cross-paradigm nature is highlighted where relevant.

### 5.1 Axis 1: Information Source and Semantic Representation

The foundational step of any graph recommendation pipeline is constructing the input topology. Traditional systems rely heavily on simple, homogeneous bipartite graphs (user-item clicks). To push the Accuracy boundary of the GNN Trilemma, recent architectures explicitly reconstruct the graph to capture rich semantic, behavioral, and relational signals before message passing even begins.

Table 2: Structural and empirical analysis of the GNN Trilemma across modern recommender architectures.

| Model | Primary Optimization | Structural Expressiveness | Theoretical Complexity | Interpretability | Main Tradeoff |
|---|---|---|---|---|---|
| **RecMind** Wang et al. (2024) | Semantic reasoning | Aligns GNN embeddings with frozen LLM priors via InfoNCE; utilizes a node-wise gating mechanism to fuse semantics and topology. | Bounded at $O(|E|d)$; primary bottlenecks shift to offline textual encoding and in-batch negative sampling. [1] | Moderate (Learned scalar gating explicitly quantifies reliance on textual semantics vs. structural graph topology). | Sacrifices full LLM fine-tuning and generative rationales to maintain inference scalability via lightweight adapters. |
| **CDRec** Liu et al. (2026a) | Continuous-time diffusion modeling | Models discrete interactions via a continuous-time SDE with popularity-aware mask scheduling and multi-hop structural guidance. | Intractable reverse transition across $|V| \times |V|$ matrices; mitigated by a deep consistency function for pseudo-single-step generation. [2] | Low–Moderate (Trajectory analysis reveals "easy-to-hard" logic, unmasking popular core interests before niche items). | Trades strict step-by-step reverse transition rate estimation for a scalable pseudo-Euler consistency approximation. |
| **DiffNBR** Zhang et al. (2026b) | Spatio-temporal diffusion | Dual DDPMs encode spatial composition and temporal evolution; Information Bottleneck decouples routine habits from exploratory intents. | $O(|U|(|S|^2 + |B|^2))$, strictly bottlenecked by the quadratic self-attention costs within both diffusion modules. [3] | Moderate (Information Bottleneck gating disentangles routine repurchases from novel, threshold-meeting exploratory items). | Sacrifices precise routine repurchase representation to heavily focus generation on uncertain, exploratory item combinations. |
| **TGODE** Fu et al. (2025) | Continuous-time temporal modeling | Decouples user time graphs and item evolution graphs via Neural ODEs; uses a diffusion generator to augment unobserved timestamps. | Heavy computational cost from iterative Runge–Kutta numerical integration combined with multi-step Markovian diffusion. | Moderate (Mathematically isolates intrinsic personalized preference drift from global, extrinsic popularity shifts). | Abandons the efficiency of discrete-time decoders (like pure Transformers) to bear the heavy numerical costs of continuous ODE solvers. |
| **LightGNN** Chen et al. (2025a) | Lightweight graph propagation | Employs an end-to-end sparse decision matrix to eliminate redundant edges; preserves signals via high-order knowledge distillation. | Massively reduces spatial/inference bounds; bottlenecks shift to the training phase which requires computing dense pairwise distillation similarities. | Low–Moderate (Learned sparse matrix explicitly identifies which noisy behavioral edges were excised from the logic). | Achieves extreme inference scalability but requires a computationally heavy, multi-stage hierarchical distillation training pipeline. |
| **LightKG** Li et al. (2025b) | Efficient knowledge-aware recommendation | Bypasses dense matrices by encoding directed knowledge graph relations as scalar pairs embedded within linear aggregation functions. | $O(d(|G_{UI}|+|G_{KG}|))$; avoids complex subgraph generation, limiting bottlenecks to full-graph node similarity contrastive calculations. [4] | Low–Moderate (Scalar magnitude explicitly quantifies the global importance of specific semantic KG edge types). | Trades fine-grained, multi-hop relational expressiveness for scalable global semantic pattern efficiency. |
| **AlphaFree** Jeon et al. (2026) | Graph-free scalable recommendation | Discards K-hop propagation; aligns MLP representations with sets of behaviorally and semantically similar items via InfoNCE. | Inference bounded to highly efficient $O(d_{LR}dn+n \log K)$; absorbs a heavy quadratic $O(n^2)$ offline preprocessing bottleneck. [5] | Low (Lacks topological paths, but semantic filtering thresholds dictate which text-based similarities form the "neighborhood"). | Absorbs massive offline $O(n^2)$ pairwise similarity computation costs to permit ultra-lightweight, purely MLP-based inference. |
| **PULSE** Choi et al. (2026) | Parameter-efficient recommendation | Replaces user embeddings with an RBF-kernel weighted fusion of broad community affiliations and specific neighbor-item interactions. | Eliminates the $O(|U|d)$ user embedding table; bottlenecks shift to multi-graph propagation and pairwise similarity tracking. [6] | Moderate (Scalar gating weights $\alpha_u$ dynamically balance reliance on global community homophily versus localized peer influence). | Sacrifices fine-grained, standalone user ID personalization in favor of a highly compressed, socially-derived representation paradigm. |
| **EAGLE** Li et al. (2025a) | Efficient temporal graph learning | Decouples signals into short-term recent neighbors and long-term Temporal Personalized PageRank (T-PPR) via exponential time-decay fusion. | Bypasses deep Transformers/GNNs; spatial and computational costs strictly bounded by the dynamic maintenance of top-$k_s$ T-PPR matrices. | Moderate (Decay mechanism explicitly isolates whether predictions were driven by immediate recency or long-term structural proximity). | Completely abandons deep, localized multi-hop expressiveness to rely entirely on lightweight MLPs operating over heuristically filtered subsets. |
| **SoREX** Guo et al. (2025) | Self-explainable recommendation | Randomly samples candidate-aware multi-hop ego-paths and integrates them back into user embeddings via attention-based re-aggregation. | Bounded at $O(L|E|d + n_w|E_r|d)$; heavy inference bottlenecks stem from real-time random walk sampling and subsequent attention fusion. [7] | High (Explicitly extracts and visualizes structural motifs like "co-purchase" or "friend-of-friend" overlapping paths). | Trades standard dense GNN inference scalability to bear the heavy computational cost of real-time explicit ego-path extraction. |
| **BEAT** Feng et al. (2026) | Disentangled explainable recommendation | Disentangles behaviors via VQ-VAE into a discrete behavior vocabulary, explicitly aligned with semantic priors from a frozen LLM. | Bottlenecks shift to offline dual-codebook quantization; real-time inference relies entirely on lightweight projectors to avoid LLM limits. | High (Maps continuous topological vectors into discrete textual tokens to prompt coherent, human-readable LLM rationales). | Abandons end-to-end LLM fine-tuning, forcing the model to reason strictly over compressed discrete tokens to maintain system scalability. |
| **TopKGAT** Chen et al. (2026b) | Objective-aligned ranking | Employs a band-pass activation function to filter extreme relevance, focusing aggregation expressiveness entirely on uncertain, borderline candidates. | Avoids global matrices; the primary bottleneck is calculating continuous gradient ascent dynamics and layer-wise threshold parameters. | Moderate (Layer-specific threshold tracking $(\beta_u^{(l)})$ exposes the hierarchical transition from generic to borderline relevance ranking). | Sacrifices traditional graph-smoothing uniformity for a strictly discriminative, boundary-focused attention mechanism. |
| **FastPFRec** Yan et al. (2026) | Federated scalable recommendation | Mitigates over-smoothing via asynchronous updates: frequent user evolution combined with intentionally frozen, sparsely updated item embeddings. | Reduces item-side convolution cost by factor of $h$; bottlenecks shift to LDP noise injection and trusted-node parameter communication. | Low (Deliberately obscures topological motifs and structural transparency to cryptographically guarantee local differential privacy). | Trades exact global graph visibility and item-side representational freshness for cryptographically secure federated training scalability. |
| **Data-Free Model Extraction** Wang et al. (2026) | Black-box model extraction | Uses a generalization-aware surrogate that freezes synthetic user embeddings, capturing platform-level latent interaction patterns. | Discards user embedding tables; bottlenecks strictly dictated by API query limits and the continuous synthesis of virtual interaction graphs. | Low (Reverse-engineering focus: translates opaque target logic into inspectable surrogate graph convolutions to expose vulnerabilities). | Sacrifices training stability on real-world distributions, relying entirely on hallucinated interactions and generalization constraints. |
| **Attack by Unlearning** Zhang et al. (2026a) | Robustness under graph unlearning | Exploits unlearning dynamics via bi-level optimization, using a surrogate GNN to pseudo-label and align injected target connections. | Dense attack $O(L(E + mn)d)$ is bottlenecked to $O(L(E + m \cdot n_{sub})d)$ by restricting structural optimization strictly to a $k$-hop candidate subgraph. [8] | Low–Moderate (Exposes the specific localized topological motifs that approximate unlearning operators inadvertently amplify into global collapse). | Sacrifices exact calculation of the full unlearning trajectory in favor of a computationally feasible, one-step gradient descent approximation. |

[1] $|E|$: Number of edges, $d$: Embedding dimension.  [2] $|V|$: Total items/discrete states in the vocabulary.  [3] $|U|$: Number of users, $|S|$: Max sequence length, $|B|$: Max basket size.
[4] $|G_{UI}|$: User-Item graph size, $|G_{KG}|$: Knowledge Graph size.  [5] $d_{LR}$: Language representation dimension, $n$: Number of items, $K$: Top-K threshold.  [6] $|U|$: Number of users.
[7] $L$: Number of layers, $n_w$: Number of random walks, $|E_r|$: Edges in interaction rating matrix.  [8] $m$: Injected malicious nodes, $n_{sub}$: Nodes in candidate sub-graph, $k$: Propagation hops.

Table 3: Architectural pressures and tradeoffs across the proposed GNN Trilemma dimensions.

| Architectural Direction | Accuracy | Scalability | Explainability | Primary Limitation |
|---|---|---|---|---|
| Continuous-time and Neural ODE Models | High | Low–Moderate | Moderate | Continuous temporal dynamics improve sequential fidelity but introduce expensive training and optimization procedures |
| Diffusion-based Recommendation Models | High | Low | Low–Moderate | Iterative denoising and multi-step sampling substantially increase computational overhead |
| Lightweight Graph Architectures | Moderate–High | High | Low | Simplified propagation may reduce expressive capacity and semantic richness |
| Graph-free / Graph-bypass Architectures | Moderate–High | Very High | Low | Depend heavily on semantic metadata and may lose structural collaborative interpretability |
| LLM-Hybrid Recommendation Systems | High | Low | High | Large semantic models introduce substantial memory, latency, and deployment complexity |
| Explainable Recommendation Architectures | Moderate–High | Low | Very High | Path extraction, rationale generation, and token-level alignment increase inference cost |
| Federated and Privacy-aware Systems | Moderate | Moderate | Low | Communication overhead and privacy constraints may limit optimization flexibility |
| Objective-aligned Recommendation Models | High | Moderate | Moderate | Explicit optimization toward fairness, diversity, or top-K objectives may reduce general adaptability |

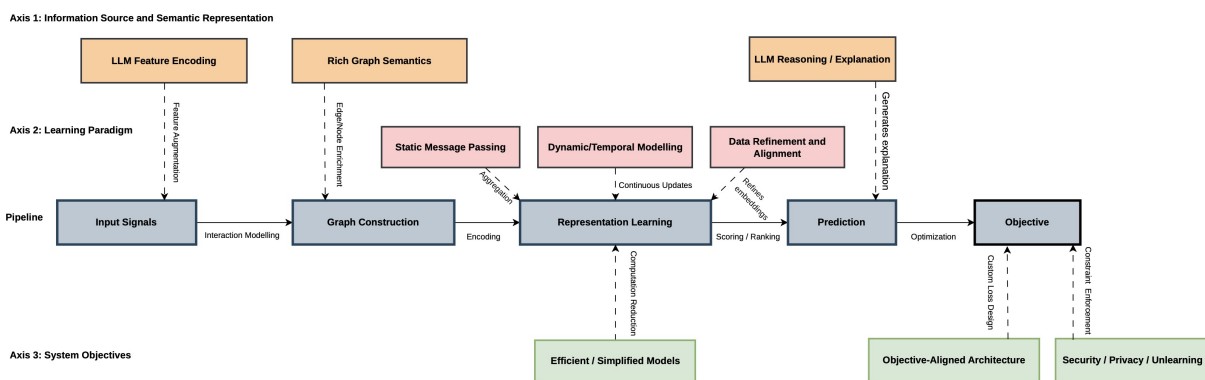

Figure 1: The Integration Pipeline of Modern GNN-Based Recommender Systems. Rather than isolated improvements, recent advancements act as modular enhancements to the core recommendation pipeline across three axes: Information Source, Learning Paradigm, and System Objectives.

### 5.1.1 Modeling Rich Graph Semantics

Modern approaches to semantic enrichment can be broadly grouped into three structural paradigms: multi-behavior modeling, knowledge graph integration, and signed/social topology.

**Multi-Behavior and Cross-Domain Signals:** In real-world platforms, users exhibit multiple types of interactions (e.g., viewing, adding to cart, purchasing). Treating all edges uniformly obscures specific user intent. To address this, SWGCN (Chen et al., 2026a) constructs separate bipartite graphs for each behavior type. Rather than utilizing standard transformer attention, SWGCN introduces a Target Preference Weigher (TPW) to compute dynamic, edge-level importance scores, allowing the model to recognize that a "viewed-and-purchased" edge sequence holds higher synergy than a random click. Similarly, RankGraph (Wu et al.,

2025b) tackles cross-domain recommendation by constructing a massive heterogeneous graph connecting users, posts, and ads via multi-hop semantic edges. It utilizes a GPU-accelerated GNN to extract structural subgraphs, which are then injected as contextual tokens into sequential foundation models. This shift toward heterogeneous graph construction is also evident in large-scale industrial recommender systems. For example, LinkedIn's cross-domain GNN framework (He et al., 2025a) unifies signals originating from multiple platform surfaces by categorizing interactions into engagement, affinity, and intrinsic relation types. By processing a graph containing hundreds of billions of heterogeneous edges through decoupled entity encoders, the system captures semantic overlap across otherwise disconnected recommendation contexts while preserving the low-latency serving requirements of real-time notification recommendation.

Recent theoretical analyses further suggest that semantic enrichment itself can alter the spectral properties of the recommendation graph. Specifically, integrating additional graph-structured side information induces a spectrum shift in the augmented adjacency matrix, causing conventional graph filters to operate on frequency regions different from those assumed during model design (He et al., 2025b). Consequently, richer semantic connectivity does not necessarily translate into better recommendations unless the resulting spectral distortion is explicitly corrected.

*Architectural Limitation:* While these methods significantly improve predictive accuracy and conversion rates, they face severe scalability penalties. SWGCN's requirement to maintain multiple adjacency matrices and compute dynamic weights per edge scales memory at $\mathcal{O}(R|E|)$. (where $R$ is the number of behaviors). Likewise, RankGraph requires complex out-of-batch negative sampling and heavy GPU acceleration, making real-time heterogeneous subgraph extraction a major engineering bottleneck.

**Knowledge and Relational Semantics:** Beyond user interactions, item metadata provides crucial context. Standard graph recommenders often discard rich textual descriptions in favor of structural IDs. Färber et al. (2025) bridge this gap by natively integrating Resource Description Framework (RDF) Knowledge Graphs into the GNN pipeline. The authors treat semantic content as first-class graph information by converting RDF datatype properties (literals, strings, descriptions) into dense vectors via BERT, fusing them with topological object properties trained via Knowledge Graph Embeddings (e.g., TransE).

*Architectural Limitation:* This creates a heavy, multi-stage pipeline. The model's accuracy becomes highly dependent on the quality of the semantic metadata; if the underlying RDF literals are sparse, noisy, or inconsistent, the semantic advantage rapidly degrades.

**Signed and Social Topology:** Edges in recommendation graphs are not strictly positive; users actively dislike items or form complex social networks. SMA-GNN (Zhao et al., 2025) introduces a symbol-aware architecture for signed link prediction. By extracting a closed $k$-hop subgraph around a target link and applying Two-Anchor Distance Labeling (TADL), the model explicitly conditions message passing on edge signs. Positive edges boost similarity, while negative interactions mathematically invert or attenuate the message vector. Taking social semantics a step further, SoREX (Guo et al., 2025) merges the user-item bipartite graph with a user-user friendship graph. It utilizes a candidate-aware ego-path extraction mechanism, re-aggregating readable social explanation paths directly back into the user embedding before final scoring.

*Architectural Limitation:* These architectures illustrate the GNN Trilemma's tension between Explainability/Accuracy and Scalability. By forcing the model to extract explicit ego-paths and perform random-walk sampling, SoREX achieves high explainability but executes one to two orders of magnitude slower than a standard LightGCN baseline. Similarly, SMA-GNN avoids global graph memory limits but suffers from severe per-example computational overhead due to $k$-hop subgraph extraction and TADL computation during training.

### 5.1.2 Hybridization and Cross-Paradigm Integration

Although graph-based architectures effectively capture collaborative structural signals, they remain inherently semantically limited, as they cannot fully exploit the rich textual and contextual information available in modern platforms. To address this limitation, recent recommender systems increasingly combine Graph

Neural Networks (GNNs) with external paradigms, particularly Large Language Models (LLMs) and conventional Deep Neural Networks (DNNs), to improve representation quality while balancing computational and deployment constraints. Recent Graph Foundation Model surveys further categorize these integrations according to how structural graph information interacts with language representations, ranging from lightweight embedding-level alignment to deeper context-level fusion mechanisms (Wu et al., 2025a). The models discussed below primarily illustrate different points along this integration spectrum.

One parameter-efficient strategy uses Large Language Models solely as offline semantic feature extractors. For example, RecMind (Wang et al., 2024) addresses the semantic limitations of collaborative filtering by integrating a LightGCN (He et al., 2020) backbone with a frozen Large Language Model. Instead of performing runtime text generation, the framework processes item descriptions offline to generate dense semantic knowledge vectors. These semantic priors are projected into the GNN latent space through lightweight Multi-Layer Perceptron (MLP) adapters and aligned using a symmetric contrastive loss. During message passing, the model dynamically combines structural graph signals with semantic representations through a learned gating mechanism. By freezing the language model and avoiding end-to-end backpropagation through the transformer, RecMind captures rich language semantics without incurring the substantial VRAM consumption and inference latency associated with autoregressive models. The reliance on offline extraction limits adaptability to emerging vocabulary, sudden cultural shifts, or rapidly evolving content unless computationally expensive reprocessing pipelines are executed.

A related direction leverages language models to enrich graph structure itself before downstream recommendation. For example, AutoGraph (Shan et al., 2025) employs large language models and vector-quantized latent factors to identify missing semantic connections and introduce auxiliary graph components that capture global preference patterns. Rather than performing semantic reasoning during recommendation, such LLM-augmented graph approaches improve the topology presented to the GNN offline, allowing richer structural information to be incorporated without incurring additional inference-time language model costs.

To establish a more interpretable connection between structural and semantic representations, BEAT (Feng et al., 2026) introduces a discrete behavior vocabulary. Instead of directly merging continuous embeddings, the framework processes the user–item graph through a Graph Convolutional Network (GCN) and applies Vector Quantized Variational Autoencoding (VQ-VAE) to compress latent user behaviors into discrete tokens. These learned tokens represent both coarse-grained preferences and fine-grained behavioral intentions, which are subsequently aligned with the semantic space of a frozen Large Language Model. This design enables the system to achieve strong structural disentanglement and expressive behavioral modeling before semantic reasoning is performed by the language model.

In contrast, CoLaKG (Cui et al., 2025b) represents a heavier form of Large Language Model integration, where the language model functions not only as an encoder but also as an active knowledge extractor and semantic reasoner. Traditional Knowledge Graph (KG)-based recommenders often suffer from incomplete attributes and depend on noisy multi-hop graph propagation to capture distant relationships. CoLaKG mitigates these limitations through a Knowledge Graph Retrieval-Augmented Generation (KG-RAG) framework. The system extracts local item-centric subgraphs, converts them into textual prompts, and queries the language model to infer missing facts and summarize semantic relationships. Additionally, globally related nodes are retrieved to supplement sparse graph regions. Although this active reasoning significantly improves semantic understanding compared with conventional Knowledge Graph-GNN pipelines, it introduces substantial computational overhead. Repeated language model inference over graph-derived textual contexts requires long prompts and extensive semantic processing, forcing the architecture to rely heavily on offline caching and distilled representations to maintain practical inference latency in production settings.

The trade-off between structural expressiveness and industrial-scale deployment is explicitly addressed by LinkSAGE (Liu et al., 2025). Operating on a massive heterogeneous job marketplace graph containing more than one billion users and fifty million jobs, the framework avoids the prohibitive cost of real-time graph inference. Instead, LinkSAGE adopts a decoupled architecture in which a GraphSAGE encoder is trained independently to generate inductive embeddings for users and jobs while leveraging skill-based graph densification to alleviate interaction sparsity. These precomputed graph embeddings are subsequently incorporated as feature augmentations within highly optimized Deep Neural Network ranking pipelines. By

Figure 2: Evolution from static to temporal GNN-based recommendation systems. **Left:** Traditional static approaches operate on discrete graph snapshots with fixed structures. **Right:** Modern temporal methods model preferences as continuous processes through (A) Neural ODEs for smooth evolution, (B) Diffusion models for denoising, and (C) Lifecycle modeling for interest decay.

relying on nearline embedding refresh instead of real-time message propagation, LinkSAGE deliberately sacrifices continuous graph dynamism in exchange for the strict serving cost, freshness, and sub-millisecond latency requirements demanded by billion-scale production systems.

## 5.2 Axis 2: Learning Paradigms

Beyond the structural composition of the input graph discussed in Axis 1, the computational approach used to model user-item interactions represents a critical architectural choice. Traditional GNN-based recommendation systems predominantly operate on discrete, static graph snapshots, utilizing standard layer-wise propagation and ranking objectives. Recent literature indicates a structural shift to address the limitations of static modeling and data sparsity. This section categorizes these evolving learning paradigms into three primary directions: continuous and dynamic preference modeling, data refinement and representation alignment, and efficient, scalable architecture design.

### 5.2.1 Continuous and Dynamic Preference Modeling

As illustrated in Figure 2, modeling temporal dynamics often requires moving beyond the discrete, layer-wise updates of traditional static graphs. In a static framework, time is artificially quantized into rigid snapshots, and message passing assumes that a user's preference remains stationary across the extracted session. However, real-world user interests decay, peak, and shift continuously. To resolve this, modern continuous-time frameworks treat the user's representation as a continuous state trajectory, $h(t)$. By leveraging sophisticated mathematical mechanisms such as Graph Neural Ordinary Differential Equations (ODEs) to model smooth preference drift, temporal diffusion to stochastically denoise fading historical interactions, and data-driven lifecycle functions to map the exact phases of an interest, these models achieve exact temporal granularity, though often at the expense of new scalability constraints.

**Continuous-Time Evolution using Neural ODEs:** To handle irregular temporal gaps, models like TGODE (Fu et al., 2025) abandon discrete propagation entirely. Instead, they define the evolution of user

and item embeddings as a continuous differential equation: $\frac{dh(t)}{dt} = f(h(t), t, \mathcal{G})$, allowing an ODE solver to compute embeddings at arbitrary, continuous timestamps. This approach is particularly effective in highly dynamic environments, such as out-of-town trip recommendation. For example, SPOT-Trip (Liu et al., 2026b) explicitly separates static long-term tastes (transferred via a Knowledge Graph) from dynamic travel intents. It utilizes a Neural ODE combined with a temporal point process to model how a user's intent continuously evolves throughout a single day of travel. However, this continuous precision comes at a direct cost to scalability. ODE solvers require iterative numerical integration (e.g., Runge-Kutta), which is computationally heavier than standard forward-pass GNNs, introducing unpredictable runtime variance and optimization instability on large-scale graphs.

**Data-Driven Lifecycle Modeling:** Rather than modeling time as a strict mathematical continuum, some architectures focus on the behavioral lifecycle of an interest. Moving away from naive exponential decay functions (e.g., $w = e^{-\lambda t}$), DILN (Cai et al., 2025) argues that user interests pass through distinct physiological phases: emergent, stable, and declining. By constructing historical activity histograms to capture growth and decline trends, DILN routes these lifecycle embeddings through a Mixture-of-Experts (MoE) architecture. This allows the model to dynamically increase the exposure of newly forming interests while proactively down-weighting fading ones, relying on data-driven phase awareness rather than strict chronological time.

**Generative Diffusion for Sequential Denoising:** The third major paradigm shift in temporal modeling is the adoption of diffusion models to refine noisy sequential interactions. Rather than adding noise to graph edges or raw timestamps, these models progressively corrupt and reconstruct the latent item representations or basket states. For instance, CDRec (Liu et al., 2026a) applies continuous-time diffusion over discrete categorical states to predict future interactions. Expanding on this, DiffNBR (Zhang et al., 2026b) targets the Next-Basket Recommendation problem by combining spatio-temporal diffusion with the Information Bottleneck principle. By mathematically maximizing the mutual information between a compressed latent state and the future basket—while discarding spurious correlations from the noisy input history—DiffNBR prevents the model from blindly memorizing accidental item combinations. This generative denoising is also highly effective for multi-modal feature purification. As demonstrated by recent multi-modal architectures (Cui et al., 2025a), conditioning the reverse diffusion process on a user's specific behavior sequence can actively strip away distracting visual or textual noise that does not correlate with the target intent (e.g., a purchase). Despite these substantial improvements in representation robustness and uncertainty modeling, diffusion frameworks conflict with strict industrial latency constraints. The requirement of 20 to 100 iterative refinement steps during the reverse denoising process creates severe computational overhead, rendering real-time serving significantly slower than traditional autoregressive or attention-based recommenders.

### 5.2.2 Data Refinement and Representation Alignment

Real-world recommendation data is inherently imperfect. Implicit feedback (e.g., clicks) contains false negatives, multi-modal features suffer from quality disparities, and user-item interactions are often highly asymmetric. To prevent models from memorizing this noise, recent architectures focus heavily on refining data and aligning representations either before or during the learning process.

The most prevalent method for handling data sparsity has traditionally been Graph Contrastive Learning (GCL), which creates multiple "views" of a graph via heuristic augmentations like random edge dropping or node masking. However, these random perturbations can compromise the graph's true semantic meaning. To resolve this, NLGCL (Xu et al., 2025) eliminates artificial graph augmentation entirely. Operating as a plug-and-play framework, it extracts contrastive signals directly from the internal layer-wise structure of the GNN. Because a node's representation naturally shifts its semantic scope as it aggregates from wider hop-radii in deeper layers, contrasting these "naturally existing neighbor layers" provides robust feature alignment. This effectively removes the memory overhead of storing augmented adjacency matrices. Alternatively, to overcome the semantic distortion caused by random augmentations, the Path-Enhanced Contrastive Learning (PECL) (Sun et al., 2026) framework shifts the contrastive focus from isolated nodes to multi-hop topologies. PECL samples rich interaction paths and applies both intra-path and inter-path contrastive objectives. This mechanism ensures the model captures subtle, high-order relational patterns rather than relying on

ambiguous semantic augmentations at the single-node level. Beyond structural refinement, models must also refine their optimization signals. Standard negative sampling randomly selects unclicked items, heavily penalizing the model if it inadvertently samples a highly relevant item the user simply has not seen yet (a "false negative"). To correct this, Dynamic Margin-based Contrastive Learning (Chen et al., 2025c) replaces static penalties with an adaptive margin derived from the model's evolving confidence in separating a user representation from positive and negative samples. By dynamically adjusting the margin according to the relative similarity scores between the user and candidate items, the framework places greater emphasis on hard negatives that are difficult to distinguish from true preferences. Moreover, it avoids excessive penalization of potentially relevant unseen items. Although NLGCL and Dynamic Margin significantly improve optimization robustness and embedding separation, they remain black-box latent vectors, offering no improvements to system explainability. Furthermore, calculating adaptive margins across all negative samples per epoch introduces new computational overhead during training.

When graphs incorporate rich side-information, data refinement must address quality disparities across different modalities (e.g., highly informative text paired with a noisy or irrelevant image). Rather than fusing modalities directly, R2MR (Tang et al., 2025) introduces a generative alignment paradigm. It utilizes a consensus-based Modality Reviewer to evaluate the quality of incoming features, and a Modality Rewriter to generatively reconstruct low-quality modalities using latent mappings from the high-quality ones. While this explicitly solves modality imbalance and yields high accuracy gains, it introduces severe model complexity. The generative rewriting process is computationally expensive at web-scale and runs the risk of hallucinating features or distorting semantics, further challenging industrial deployment.

Finally, the community has begun utilizing generative diffusion not just for sequential temporal denoising (as seen with DiffNBR (Zhang et al., 2026b) and CDRec (Liu et al., 2026a) in Section 5.2.1), but as a highly principled engine for robust contrastive augmentation. Standard graph augmentations, such as random edge dropping, often distort critical structural signals and degrade semantic consistency across augmented views. To resolve this, frameworks like RaDAR (Huang et al., 2026) replace random topological noise with a dual-view generation architecture. RaDAR utilizes a variational autoencoder-based graph generative model to capture global structural semantics and a relation-aware denoising model to adaptively reweight edge contributions. Rather than corrupting the graph structure, RaDAR applies a latent diffusion paradigm directly to the GCN-derived node embeddings. By progressively injecting Gaussian noise in the latent space and pairing it with an asymmetric contrastive learning objective, RaDAR maintains semantic consistency and aligns structurally similar nodes across multi-hop relations without relying on spurious patterns. However, this sophisticated approach pushes the limits of the GNN Trilemma; the fusion of generative reconstruction, relation-aware denoising, and diffusion-based contrastive learning achieves highly competitive robustness under extreme sparsity, but introduces hierarchical optimization complexity that challenges seamless deployment on large-scale industrial graphs.

## 5.3 Axis 3: System Objectives

While Axes 1 and 2 focus on maximizing predictive accuracy through enriched representations and advanced propagation strategies, real-world recommendation systems cannot operate in a vacuum. Deployment at scale introduces rigid non-functional constraints that accuracy metrics fail to capture. Axis 3 categorizes architectures based on their primary system-level optimization objectives. Rather than solely optimizing for standard ranking metrics (e.g., NDCG or Recall), the frameworks in this category fundamentally restructure the GNN pipeline to address three distinct deployment challenges: satisfying strict industrial latency and memory constraints (Efficiency and Scalability), protecting user topologies against adversarial attacks and privacy leaks (Security and Unlearning), and promoting equitable, unbiased exposure across diverse user and item groups (Objective-Aligned Design).

### 5.3.1 Efficient and Scalable Architecture Design

The profound structural complexity introduced by temporal diffusion, LLM adapters, and contrastive learning fundamentally conflicts with the strict latency and memory constraints of industrial deployment. Standard GNNs expend significant compute power passing messages across millions of redundant edges and

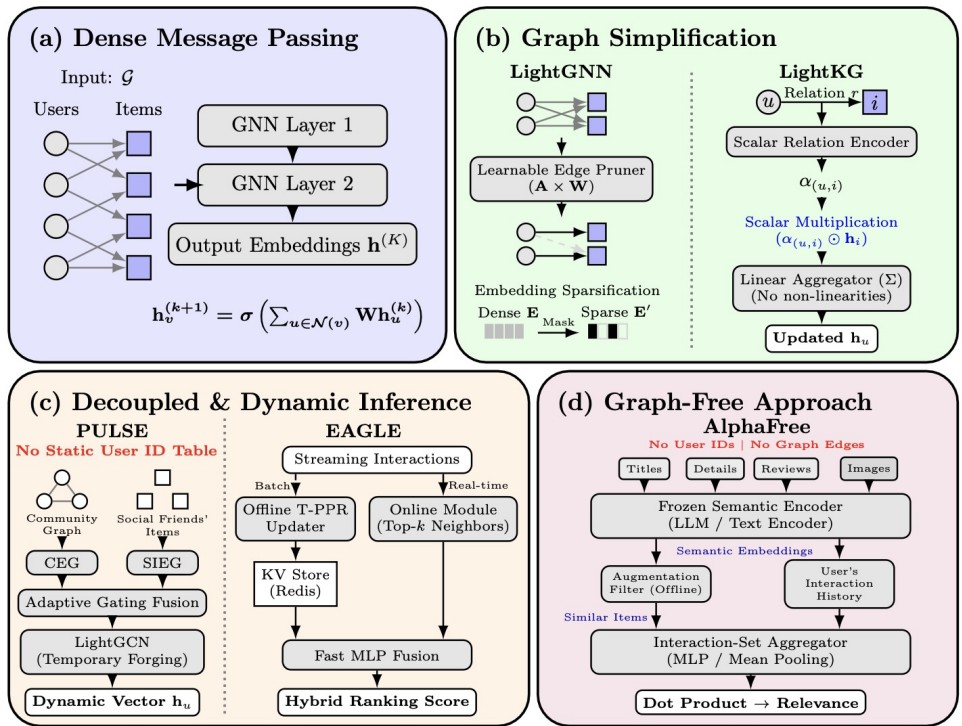

Figure 3: Architectural strategies for resolving the GNN scalability bottleneck. Compared to dense message passing (a), modern systems achieve extreme efficiency through (b) aggressive graph and embedding simplification, (c) decoupled offline/online inference pipelines, and (d) graph-free semantic representation.

storing unnecessary embedding parameters (Figure 3a). A distinct paradigm of research has emerged focused entirely on structural pruning and decoupled inference, accepting minor sacrifices in expressive power to achieve substantial gains in scalability.

**Graph and Parameter Simplification:**   The most direct approach to scalability involves aggressive pruning of the topological and parameter footprints of the model (Figure 3b). LightGNN (Chen et al., 2025a) functions as a distillation-based pruning framework. It actively learns which graph edges contribute minimally to the collaborative signal, eliminating up to 80% of redundant edges and reducing embedding entry dimensions by 90%. This prevents the model from wasting valuable memory on noise. Similarly, LightKG (Li et al., 2025b) argues that attention-heavy Knowledge Graph recommenders are unnecessarily complicated. It explicitly removes dense relation embeddings and complex attention aggregators, choosing instead to encode directed relations as lightweight scalar pairs. Although these simplifications yield an 84% reduction in training time, they inherently trigger the Trilemma's accuracy penalty. Aggressive edge removal degrades multi-hop collaborative signals and severs the long-tail interactions crucial for niche recommendations, while scalar relations severely limit the depth of semantic reasoning.

**Decoupled and Dynamic Inference:**   To avoid the online cost of deep message passing, several architectures decouple the representation generation from the static embedding table entirely (Figure 3c). Traditional recommenders assign a persistent vector $\mathbf{h}_u \in \mathbb{R}^d$ to every user, resulting in excessively large, static embedding tables that dominate memory usage. PULSE (Choi et al., 2026) eliminates explicit user embeddings. Instead, it utilizes a Dynamic Aggregator to generate temporary user representations on-the-fly, pulling signals directly from social neighbors and interacted items, yielding up to a 50% reduction in parameters.

This decoupling strategy extends to temporal processing as well. Architectures like Lighter-X (Zheng et al., 2025) separate feature transformation from graph propagation, precomputing propagation to reduce repeated

neighborhood aggregation. EAGLE (Li et al., 2025a) takes this further by splitting computation into a heavy offline structural module (using Temporal Personalized PageRank to capture long-term global influence) and a lightweight online module that only aggregates a node's most recent temporal neighbors. By replacing deep online inference with cached offline structural priors, EAGLE achieves an explicit 50× speedup over traditional transformer-based Temporal Graph Neural Networks (T-GNNs). The accepted tradeoff is reduced adaptive long-range reasoning compared to fully online temporal models.

**Graph-Free Semantic Representation:** A significant departure from traditional GNN-based recommendation is the complete removal of graph topology during inference (Figure 3d). Pushing further architectural simplification, AlphaFree (Jeon et al., 2026) completely eliminates GNN message passing, explicit user embeddings, and item ID representations. Instead, it relies on precomputed semantic Language Representations (LRs) extracted offline from item content and augments user behavior with behaviorally similar items retrieved from the semantic space. During online inference, AlphaFree bypasses graph propagation altogether and employs a lightweight Multi-Layer Perceptron (MLP) to aggregate a user's recent interaction history on-the-fly, generating recommendations without maintaining large embedding tables. This design substantially reduces online memory requirements, achieving up to a 69% reduction in GPU memory usage while also improving cold-start recommendation performance. However, these gains come at the cost of a computationally intensive offline preprocessing stage that requires pairwise item-similarity estimation, introducing an $O(n^2)$ complexity bottleneck, where $n$ denotes the number of items in the dataset. This exemplifies the scalability–expressiveness trade-off by shifting computation from online graph reasoning to offline semantic processing.

### 5.3.2 Security, Privacy, and Unlearning

As recommendation systems increasingly process highly sensitive behavioral data, they become prime targets for privacy leaks and adversarial exploitation. While optimizing for accuracy and scalability, modern architectures must also implement robust defenses, which inherently introduces new computational and performance trade-offs within the GNN pipeline.

**Data-Free Model Extraction:** A critical vulnerability in modern architectures is the susceptibility to model stealing, even when attackers have no access to the original training data or internal weights. Recent research on Data-Free Black-box Graph Model Extraction (Wang et al., 2026) demonstrates that an attacker can build a highly accurate surrogate model relying solely on API query outputs. By utilizing an interaction generator to create a synthetic user-item graph, the attacker queries the target model and uses the resulting ranked scores as pseudo-labels. A surrogate GNN (e.g., LightGCN) is then trained to mimic the victim model. GCNs are particularly effective for this exploit because they naturally capture the collaborative signals and neighborhood smoothness leaked by the target system's predictions. The primary limitations of this attack are the high cost of repeated API queries and the difficulty of perfectly reconstructing complex, nonlinear business logic.

**Privacy Preservation through Federated Learning:** To prevent raw interaction data from ever reaching a central server, systems frequently adopt Federated Learning (FL), a distributed training paradigm in which user devices collaboratively train a shared model without transmitting raw interaction data. Traditional federated GNNs are computationally slow due to expensive graph propagation and severe communication overhead. FastPFRec (Yan et al., 2026) addresses this by decoupling the network update schedule. Recognizing that user preferences change rapidly while item attributes remain relatively stable, the framework updates user embeddings frequently while refreshing item embeddings only periodically. To preserve privacy, it employs Local Differential Privacy (LDP), which perturbs local model updates with calibrated noise before transmission, reducing the risk of exposing individual user behavior. Furthermore, FastPFRec introduces a three-tier federated architecture with trusted intermediate nodes that securely aggregate client updates and help detect anomalous contributions, reducing the risk of malicious parameter injection while maintaining efficient convergence. While FL protects privacy, it often masks severe performance degradation on disadvantaged node groups. Frameworks like BoostFGL (Chen et al., 2026c) directly address this by introducing client-side topology boosting to correct label skew and confounding message propagation. This

ensures that privacy mechanisms do not disproportionately harm minority nodes. The inherent Trilemma trade-off in these systems is that adding LDP noise and enforcing federated boundaries predictably degrades raw recommendation accuracy compared to centralized training.

**Machine Unlearning and Adversarial Exploits:** With the enforcement of data privacy regulations (e.g., GDPR), systems must legally honor user deletion requests, often referred to as the "Right to be Forgotten." In a GNN, information propagates through topological neighborhoods, meaning a single edge deletion leaves residual influence across many embeddings. Because retraining a massive graph from scratch is computationally infeasible, novel unlearning paradigms selectively remove a user's data. For instance, UnlearnRec (Chen et al., 2025b) introduces a pre-training paradigm where an Influence Encoder estimates the spillover effect of unusable data, directly outputting updated parameters with minimal fine-tuning. Alternatively, frameworks like D2DGN (Distill to Delete) (Sinha et al., 2026) utilize knowledge distillation—employing a pretrained "preserver" to retain safe knowledge and a "destroyer" to isolate targeted deletion—achieving efficient, model-agnostic unlearning without calculating complex mathematical Hessian approximations.

However, this necessary compliance mechanism introduces a novel attack surface. The *Attack by Unlearning* framework (Zhang et al., 2026a) demonstrates how deletion requests can be weaponized. An attacker injects carefully crafted, seemingly harmless malicious nodes into the graph during the training phase. Once the model is deployed, the attacker requests the deletion of those specific nodes. Because the system relies on approximate unlearning rather than exact retraining, the localized parameter repair process mathematically destabilizes the surrounding neighborhood, collapsing accuracy and corrupting graph consistency. This reveals a significant structural vulnerability: the exact mathematical mechanisms designed to protect user privacy can be exploited to compromise the model's integrity.

### 5.3.3 Objective-Aligned Architecture Design

The final category of our taxonomy encompasses architectures where the fundamental structure of the neural network is explicitly engineered to match a specific deployment goal. Rather than relying on standard message passing to achieve broad, generic accuracy, these models embed human-centric or business-logic objectives directly into their mathematical frameworks.

**Alignment for Ranking Objectives:** Standard GNN recommenders are traditionally optimized using generic classification or point-wise loss functions, which do not structurally align with the real-world goal of retrieving the top few relevant items. TopKGAT (Chen et al., 2026b) addresses this mathematical disconnect by explicitly deriving its architecture from a differentiable approximation of Precision@K. Instead of utilizing standard attention mechanisms that learn generic neighbor importance, TopKGAT's aggregation coefficients are driven by the gradient ascent dynamics of Top-K metrics. This forces the attention weights to prioritize neighbors that specifically improve an item's ranking position. While TopKGAT efficiently aligns message passing with ranking objectives, increasing architectural depth introduces additional challenges. Deeper configurations are more vulnerable to over-smoothing and noise accumulation, and the need to learn a distinct threshold parameter ($\beta$) at every layer increases optimization complexity, making training progressively harder to stabilize.

**Alignment for Diversity and Balance:** In commercial video game recommendations, platforms must balance suggesting highly popular items (accuracy) with exposing players to niche, long-tail titles (diversity). Traditional GNNs naturally favor popular items due to their high degree of connectivity, which leads to over-smoothing and homogenized recommendations. CPGRec+ (Li et al., 2026) counteracts this by replacing standard binary edges with a Preference-Informed Edge Reweighting module. By analyzing normalized playtime and average ratings, the model assigns signed and weighted edges that distinguish genuine personal interest from generic popularity. It incorporates Large Language Models (LLMs) to generate semantic, preference-reasoned representations. By explicitly modeling both preference strength and semantic similarity, CPGRec+ demonstrates that improving diversity does not necessarily require sacrificing recommendation accuracy. However, the framework depends heavily on the availability of rich behavioral metadata, such as playtime and rating information, to construct meaningful edge weights. Although the LLM-derived

representations are computed offline, real-time inference remains dominated by graph operations on the large player–game interaction graph.

**Alignment for Fairness and Debiasing:** Because GNN aggregation inherently amplifies popularity bias (the "rich-get-richer" effect), objective-aligned design frequently targets fairness. While many frameworks attempt to fix this by altering the training loss, recent research proposes direct embedding-space correction. The Post-hoc Popularity Bias Correction framework (Islam et al., 2026) identifies that bias is already embedded in the learned representations. Without retraining the GNN, the method estimates a "popularity direction vector" within the latent space and mathematically projects user and item embeddings away from it. By explicitly removing the popularity component, the model enforces a fairer distribution of recommendations. The effectiveness of this post-hoc correction depends on careful tuning of dataset-specific hyperparameters, particularly the preference centroid coefficient ($\phi$) and the popularity penalty ($\beta$). In datasets with severe popularity skew, estimating the popularity direction reliably becomes more challenging, which can reduce the effectiveness of the debiasing process.

**Cross-Paradigm Alignment:** The philosophy of objective-aligned design extends across the entire recommender pipeline. As discussed in previous sections, architectures can be explicitly wired to prioritize explainability by extracting readable ego-paths (SoREX (Guo et al., 2025)), to achieve negative sampling robustness by computing dynamic margins based on representation similarity (Dynamic Margin (Chen et al., 2025c)), or to capture temporal adaptation by mapping the exact lifecycle phases of a user's interest (DILN (Cai et al., 2025)). Together, these frameworks demonstrate that the future of GNN recommendation relies not just on learning from graphs, but on actively shaping the learning process to meet strict, real-world deployment goals.

**Taxonomy Summary**

To synthesize the diverse architectural strategies discussed throughout this section, Table 4 provides a comprehensive overview of our proposed taxonomy. By categorizing recent literature across information sources (Axis 1), learning paradigms (Axis 2), and system objectives (Axis 3), the table highlights the dominant technical directions and representative frameworks driving the field forward. While these advancements demonstrate remarkable theoretical sophistication in modeling complex topologies and enforcing system constraints, their practical utility relies heavily on how they are empirically validated. Accordingly, the next section examines current evaluation methodologies for GNN-based recommender systems and discusses several emerging challenges in aligning experimental settings with real-world deployment conditions.

## 6 The Evaluation Crisis: Disconnect Between Architecture and Reality

As detailed in Section 5, the field of GNN-based recommendation has experienced rapid architectural diversification, adopting continuous-time differential equations (e.g., TGODE (Fu et al., 2025)), generative diffusion (e.g., DiffNBR (Zhang et al., 2026b)), and LLM-driven semantic reasoning (e.g., RecMind (Wang et al., 2024)). However, as illustrated by the dataset analysis in Table 5, the methodologies used to evaluate these sophisticated architectures have not evolved at the same pace. This section critically examines the growing gap between modern model complexity and traditional evaluation protocols.

### 6.1 The Static Dataset Trap

Table 5 highlights an overwhelming dependence on static, offline benchmark datasets across modern research. The vast majority of works continue to evaluate performance using historical datasets such as Yelp, Gowalla, Amazon-Book, and MovieLens-1M. While these datasets were instrumental in benchmarking early collaborative filtering models, they are often insufficient for comprehensively evaluating modern temporal and dynamic architectures. Recent surveys of industrial recommender systems have confirmed that widely used benchmarks such as MovieLens and Amazon largely exhibit stationary interaction patterns and comparatively mild cold-start conditions, which differ substantially from the evolving distributions observed in production environments (Zou & Sun, 2025).

Table 4: Representative methods across the proposed taxonomy of modern GNN-based recommender systems.

| Taxonomy Category | Representative Papers | Key Technical Direction | Typical Recommendation Setting | Dominant System Objective |
|---|---|---|---|---|
| **Modeling Rich Graph Semantics 5.1.1)** | SWGCN, SMA-GNN, RankGraph, Bridging RDF KG with GNNs | Modeling heterogeneous, signed, and multi-behavior interaction semantics beyond binary user–item edges through relation-aware aggregation and structured knowledge integration | Multi-behavior recommendation, heterogeneous recommendation, and knowledge-aware recommendation systems | Improving semantic expressiveness and representation fidelity |
| **Hybridization & Cross-Paradigm Integration 5.1.2)** | RecMind, BEAT, Automatic Graph Construction using LLMs, Comprehending KGs with LLMs | Integrating GNNs with LLM-driven semantic reasoning, graph construction, cross-modal alignment, and explainable recommendation mechanisms | Explainable recommendation, semantic recommendation, and knowledge-enhanced recommendation | Bridging collaborative graph signals with semantic reasoning capabilities |
| **Continuous and Dynamic Preference Modeling 5.2.1)** | Time Matters, Continuous-time Discrete-space Diffusion, Dynamic Time-aware Continual Learning, Interest Changes | Modeling user preference evolution through continuous-time dynamics, temporal diffusion, lifecycle-aware adaptation, and continual representation updating | Sequential recommendation, temporal recommendation, and evolving user-interest modeling | Capturing temporally adaptive and continuously evolving user preferences |
| **Data Refinement and Representation Alignment 5.2.2)** | NLGCL, PECL, R2MR, RaDAR, Dynamic Margin Contrastive Learning | Refining noisy interaction data using diffusion-based denoising, contrastive representation alignment, robust negative sampling, and multi-view consistency learning | Sparse recommendation, noisy interaction graphs, and cold-start recommendation | Improving robustness and representation consistency under imperfect interaction data |
| **Efficient and Scalable Architecture Design 5.3.1)** | LightKG, LightGNN, AlphaFree, PULSE, Lighter-X | Reducing graph propagation overhead through lightweight architectures, embedding-efficient learning, graph-bypass mechanisms, and scalable industrial serving pipelines | Industrial-scale recommendation, billion-scale graphs, and low-latency inference systems | Balancing recommendation quality with scalability and computational efficiency |
| **Security, Privacy, and Unlearning 5.3.2)** | Data-Free Model Extraction, Recommendation Unlearning, FastPFRec, Attack by Unlearning | Analyzing privacy risks, adversarial vulnerabilities, federated graph learning, and efficient removal of user influence from trained graph models | Privacy-sensitive recommendation and federated recommendation environments | Ensuring security, privacy preservation, and controllable model behavior |
| **Objective-Aligned Architecture Design 5.3.3)** | TopKGAT, Post-hoc Popularity Bias Correction, SoREX, Interest Changes | Designing recommendation objectives and architectural mechanisms explicitly aligned with fairness, explainability, diversity, and debiasing goals | Human-centric recommendation and objective-aware ranking systems | Aligning recommendation behavior with application-level and user-centric objectives |

For instance, architectures that aim to model continuous user preference evolution are frequently evaluated on artificially quantized snapshots of static data. Even when temporal evaluation is attempted, the underlying datasets often lack the rich streaming interaction fidelity observed in real-world environments. The field is optimizing highly dynamic algorithms against largely stationary targets, raising concerns about whether incremental gains on static benchmarks reliably translate to improved performance in evolving recommendation environments.

## 6.2 The Academic vs. Industrial Divide

A second major observation is the significant gap between academic benchmarking and production-scale reality. Table 5 highlights that most academic models rely predominantly on offline ranking metrics such as Recall@K and NDCG@K. Although offline ranking metrics remain valuable for controlled comparison and reproducibility, they provide only a partial view of deployment effectiveness. Recent reviews of graph-based recommender systems have highlighted that offline evaluation protocols rarely capture evolving user contexts, feedback loops, or distribution shifts, contributing to a persistent gap between offline benchmark performance and real-world deployment quality (Markchom et al., 2025). Crucially, these metrics do not capture the constraints associated with the GNN Trilemma (Section 4). Offline benchmarks do not penalize models for requiring substantial VRAM, suffering from high inference latency, or relying on computationally prohibitive unlearning mechanisms.

In contrast, industrial models in our taxonomy, such as LinkSAGE (Liu et al., 2025) and RankGraph (Wu et al., 2025b), often bypass standard offline benchmarking entirely. Evaluated on billion-scale, continuously evolving heterogeneous graphs, these systems measure success through online A/B testing, real-time Click-

Through Rate (CTR), and conversion-oriented metrics in near-line deployment settings. This discrepancy suggests that academia largely optimizes for offline ranking precision, whereas industry must simultaneously balance precision, scalability, latency, and online engagement.

### 6.3 Pathways Toward Realistic Evaluation

To bridge this gap and improve the practical relevance of future GNN recommenders, the community must rethink existing evaluation standards. We propose four important shifts in benchmarking protocols:

- **Hardware-Aware Reporting:** Papers introducing computationally intensive mechanisms, such as diffusion denoising or LLM hybridization, should report inference latency (e.g., in milliseconds), peak VRAM utilization, training throughput, and scalability limits alongside standard accuracy metrics. Recent studies show that several GNN recommenders encounter Out-of-Memory (OOM) failures or exceed practical training budgets when evaluated on large-scale graphs (Li et al., 2025a; Chen et al., 2026c). Therefore, reported accuracy gains should be contextualized by their computational cost, including the largest graph scale that can be processed without hardware failure.

- **Standardized Temporal Benchmarks:** The field requires a transition toward streaming datasets that preserve exact interaction timestamps, item lifecycle decay, and popularity drift. This would encourage evaluation protocols based on strict chronological splits rather than randomized train-test partitions.

- **Off-Policy Evaluation (OPE):** Since academic researchers rarely have access to production-scale A/B testing environments, Off-Policy Evaluation (OPE) offers a promising intermediate framework for estimating deployment-time behavior under counterfactual interaction data. Although OPE introduces its own challenges, including exposure bias and estimator variance, it may provide a more realistic approximation of online engagement dynamics than static ranking metrics alone.

- **Robust Reproducibility Protocols:** Future evaluation standards should encourage reporting across multiple random seeds, statistical significance testing, and sensitivity analyses for hyperparameter initialization. Without reporting experimental variance, it becomes difficult to determine whether observed performance differences are statistically meaningful or consistently reproducible across runs.

Table 5: Dataset Characteristics and Evaluation Settings Across Recent GNN-based Recommendation Systems. The table highlights the dominance of static offline benchmarks, limited temporal evaluation, and the scarcity of deployment-oriented validation despite increasing architectural complexity.

| Paper | Datasets Used | Dataset Characteristics | Evaluation Metrics | Temporal / Dynamic Evaluation | Deployment Setting |
|---|---|---|---|---|---|
| TGODE Fu et al. (2025) | Amazon Beauty, Amazon Sports, Amazon Toys, Amazon Video, ML-100K | Sequential recommendation datasets with irregular interaction intervals, temporal sparsity, and popularity drift | Recall@5/10/20, MRR@5/10/20, NDCG@5/10/20 | Chronological Offline Split | Offline Benchmark Only |
| CDRec Liu et al. (2026a) | MovieLens-1M, Ciao, Dianping | Continuous-time recommendation datasets with varying interaction densities and evolving user preferences | Recall@5/10, NDCG@5/10 | Chronological Offline Split | Offline Benchmark Only |

Table 5 – continued from previous page

| Paper | Datasets Used | Dataset Characteristics | Evaluation Metrics | Temporal / Dynamic Evaluation | Deployment Setting |
|---|---|---|---|---|---|
| DiffNBR Zhang et al. (2026b) | Dunnhumby, Instacart, RetailRocket, ValuedShopper | Next-basket recommendation datasets with sequential purchases and long-term shopping patterns | Recall@5/10/20, NDCG@5/10/20 | Sequential Offline Evaluation | Offline Benchmark Only |
| BEAT Feng et al. (2026) | Amazon, Yelp, Google | Interaction and review datasets evaluated under zero-shot explainable recommendation settings | BLEU, BARTScore, BERTScore | Static Snapshot Evaluation | Offline Benchmark Only |
| AlphaFree Jeon et al. (2026) | Amazon Reviews, Steam | Recommendation datasets with textual metadata and varying interaction densities for cold-start evaluation | Recall@20, NDCG@20 | Static Random Split | Offline Benchmark Only |
| LightGNN Chen et al. (2025a) | Gowalla, Yelp, Amazon | Highly sparse collaborative filtering datasets with implicit feedback interactions | Recall@20/40, NDCG@20/40 | Static Random Split | Offline Benchmark Only |
| FastPFRec Yan et al. (2026) | Yelp, Kindle, Gowalla-100K, Gowalla-1M | Federated recommendation datasets with heterogeneous feedback and varying sparsity levels | HR@10, NDCG@10 | Static Random Split | Offline Federated Simulation |
| SoREX Guo et al. (2025) | Yelp, Flickr, Ciao, LastFM | Social recommendation datasets containing interaction and trust-network relations | HR@10, NDCG@10, Fidelity Score | Static Random Split | Offline Benchmark Only |
| TopKGAT Chen et al. (2026b) | Ali-Display, Epinions, Food, Gowalla | Advertising, review, and location-based recommendation datasets optimized for top-K ranking | Recall@20, NDCG@20 | Static Random Split | Offline Benchmark Only |
| SWGCN Chen et al. (2026a) | Taobao, IJCAI, Beibei | Multi-behavior e-commerce datasets containing views, favorites, carts, and purchases | HR@10/20/50, NDCG@10/20 | Last-Interaction Holdout | Offline Benchmark Only |
| CPGRec+ Li et al. (2026) | Steam I, Steam II | Long-tail game recommendation datasets with ratings, playtime, and rich game metadata | Recall@K, NDCG@K, Hit@K, Precision@K, Coverage Metrics | Historical Offline Evaluation | Offline Benchmark Only |
| RankGraph Wu et al. (2025b) | Industrial heterogeneous recommendation graph (Meta) | Large-scale dynamic heterogeneous graph supporting cross-domain recommendation and retrieval | Recall@K, Engagement Recall, CTR, Conversion Rate | Streaming / Online Evaluation | Industrial Online Validation |
| LinkSAGE Liu et al. (2025) | LinkedIn production graph | Billion-scale dynamic job recommendation graph with rapidly evolving user-job interactions | Online engagement and job-matching metrics | Streaming / Online Evaluation | Production Deployment + A/B Testing |

Table 5 – continued from previous page

| Paper | Datasets Used | Dataset Characteristics | Evaluation Metrics | Temporal / Dynamic Evaluation | Deployment Setting |
|---|---|---|---|---|---|
| EAGLE Li et al. (2025a) | Contacts, LastFM, Wikipedia, Reddit, AskUbuntu, SuperUser, Wiki-Talk, Trade, Genre, Token | Continuous-time graph benchmarks spanning social, communication, and transaction networks | AP, MRR, HR@10 | Chronological Offline Split | Offline Temporal Benchmark Evaluation |

*Note:* Static Random Split denotes evaluation on randomly partitioned historical interactions without preserving temporal order. Chronological Offline Split preserves interaction ordering but evaluates on fixed historical snapshots. Sequential Offline Evaluation and Last-Interaction Holdout exploit interaction sequences while remaining offline protocols. Streaming / Online Evaluation refers to continuously updated assessment under evolving user behavior and interaction distributions.

# 7 Open Challenges and Emerging Research Directions

While recent GNN-based recommendation architectures have achieved remarkable theoretical sophistication, the transition from academic benchmarking to robust industrial deployment remains constrained by significant structural and computational limitations. Based on the limitations and challenges identified throughout this survey, we outline three critical research directions for the future of GNN-based recommendation.

## 7.1 Escaping the GNN Trilemma via Lightweight Deployment

The fundamental tension between expressiveness and efficiency remains a central challenge. As demonstrated by frameworks like DiffNBR (Zhang et al., 2026b) and Time Matters (Fu et al., 2025), integrating generative diffusion or Neural ODEs improves sequential recommendation accuracy but introduces substantial computational overhead. The iterative reverse denoising processes and continuous-time integrations substantially limit their practicality in latency-sensitive environments.

Future research must focus on bridging this gap through advanced model compression and decoupled inference. Promising directions include teacher-student knowledge distillation (where a heavyweight model trains a lightweight GCN for serving) and architectural decoupling (as explored in decoupled inference frameworks such as EAGLE (Li et al., 2025a)), where expensive temporal propagation is handled asynchronously offline. Addressing the aggressive pruning trade-offs observed in LightGNN (Chen et al., 2025a) without sacrificing critical long-tail collaborative signals will be essential for real-time deployment.

## 7.2 Continual Learning on Streaming Interaction Graphs

Many current architectures still rely on periodic retraining over static graph snapshots, making adaptation to rapidly evolving interaction streams computationally expensive. While industrial systems like RankGraph (Wu et al., 2025b) demonstrate strong multi-domain capability, maintaining these heterogeneous graphs under frequent embedding updates introduces substantial preprocessing and synchronization overhead.

Recent continual recommendation research has further highlighted that adaptation is not solely a matter of incorporating new interactions. As new items continuously emerge, the underlying data distribution evolves over time, making previously learned representations progressively less reliable. Frameworks such as DITTO (Choi et al., 2025) emphasize the dual challenge of retaining useful historical knowledge while simultaneously adapting to shifting item distributions, thereby mitigating catastrophic forgetting without sacrificing responsiveness to new recommendation patterns.

Addressing these challenges will require Continual Graph Learning paradigms capable of ingesting streaming edge updates while maintaining stable and adaptable representations over time. Research must focus on preventing catastrophic forgetting while adapting to temporal preference drift and evolving item semantics. Addressing the cold-start limitations of ID-based user embeddings through ID-free, context-driven behavioral tokenization (such as the AlphaFree framework (Jeon et al., 2026)) offers a viable strategy for maintaining robustness on highly dynamic user bases.

## 7.3 Verifiable Security and Trustworthy Unlearning

As global data protection regulations (e.g., GDPR, DPDPA) mandate strict data compliance, recommendation systems must legally support the right to be forgotten. However, current approximate unlearning algorithms may introduce complex structural vulnerabilities. The *Attack by Unlearning* paradigm (Zhang et al., 2026a) demonstrates that approximate localized parameter repair may unintentionally introduce vulnerabilities that degrade model robustness or graph consistency under adversarial conditions. Future research must move beyond approximate heuristics toward formal guarantees regarding the effectiveness

and verifiability of unlearning procedures, ensuring that privacy-preserving mechanisms do not compromise the integrity of the recommendation framework.

Collectively, these challenges indicate that the future of graph-based recommendation will depend not only on improving predictive accuracy but also on developing architectures that are computationally efficient, temporally adaptive, semantically grounded, privacy-aware, and robust under real-world deployment constraints. Bridging the gap between theoretical sophistication and practical robustness remains the central challenge for the next generation of recommender systems.

# 8 Conclusion

Graph Neural Networks have fundamentally transformed the landscape of recommender systems, evolving from simple message-passing algorithms into highly sophisticated, multi-paradigm architectures. As detailed in our proposed taxonomy, modern GNN recommenders no longer merely aggregate user-item interactions; they integrate generative diffusion for temporal denoising, continuous-time differential equations for preference evolution, and Large Language Models for deep semantic reasoning. Furthermore, the field has increasingly recognized that recommendation is not solely an accuracy optimization problem, leading to the development of objective-aligned architectures that structurally enforce fairness, diversity, and privacy preservation.

As our survey highlights, this rapid theoretical advancement has created a structural and methodological crisis. The pursuit of extreme predictive accuracy has exacerbated the GNN Trilemma (Section 4), where the computational overhead of deep propagation and generative hybridization frequently renders these models infeasible for real-time, industrial deployment. This disconnect is further masked by an evaluation ecosystem that heavily relies on static, offline benchmark datasets and historical ranking metrics. As academia continues to optimize heavy, continuous-time models against stationary targets, a significant gap has emerged between academic benchmarking and the strict latency, scalability, and continual learning requirements of production-scale systems.

The next era of GNN-based recommendation must be defined by a shift from pure architectural complexity toward deployment-aware robustness. Bridging the gap between theory and practice will require the community to prioritize decoupled inference pipelines, continual graph learning for streaming interactions, and verifiable machine unlearning. By realigning our evaluation methodologies and embracing the constraints of industrial scale, the research community can ensure that the next generation of GNN recommenders is not only theoretically profound, but practically indispensable.

# Broader Impact Statement

Recommender systems significantly influence digital engagement and user behavior. Opaque algorithms and static evaluation protocols can inadvertently propagate popularity bias and compromise privacy. This survey does not introduce new algorithmic risks but instead highlights the structural tensions (captured by the GNN Trilemma) that currently constrain deployment of privacy-preserving, debiased, and transparent recommender architectures. By documenting these challenges, we aim to guide researchers and practitioners toward developing more equitable, verifiable, and privacy-compliant systems.

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
