# OpenReview forum: "The GNN Trilemma in Recommender Systems: A Survey"
_TMLR — Under review for TMLR_

### Review · Reviewer_es1s · 2026-06-17

**Summary Of Contributions:**

This survey examines the architectural evolution of GNN-based recommender systems from 2024 to 2026, making three primary contributions. First, the authors introduce the GNN Trilemma, a structured framework that formalizes the inherent tension among accuracy, scalability, and explainability in modern graph recommenders, drawing an analogy to the CAP theorem in distributed systems. Second, they propose an orthogonal three-axis taxonomy (Information Source, Learning Paradigm, System Objectives) that systematically categorizes over 30 recent architectures. Third, they diagnose an evaluation crisis, arguing that static offline benchmarks have failed to keep pace with the complexity of modern architectures.

Key strengths: The Trilemma framework is genuinely useful as an analytical lens and is applied consistently throughout the paper. The taxonomy is comprehensive and well-motivated. The coverage of recent work, spanning diffusion-based methods, LLM-hybrid systems, federated learning, and machine unlearning, is thorough and up-to-date. Tables 2 and 3 are particularly valuable as concise reference artifacts for practitioners. The evaluation crisis section raises important, underappreciated concerns about the gap between academic benchmarking and industrial deployment.

Key weaknesses: The Trilemma, while compelling as an organizing metaphor, lacks the formal rigor of the CAP theorem it is compared to (no impossibility proof or formal bound is offered). Some taxonomy assignments feel subjective given that many architectures span multiple axes. The survey's coverage skews heavily toward accuracy-oriented systems; fairness and privacy-aware work receives comparatively less depth.

**Audience:**

Yes

**Audience Explanation:**

The paper addresses one of the most active areas in machine learning research. GNN-based recommendation sits at the intersection of graph learning, generative modeling, LLM integration, and systems engineering, all topics of strong interest to TMLR's readership. The Trilemma framework and three-axis taxonomy provide vocabulary and structure that researchers and practitioners will find immediately useful for positioning new work. The evaluation crisis discussion is timely and raises methodological concerns that extend beyond recommender systems to graph learning evaluation more broadly.

**Broader Impact Concerns:**

The paper includes a Broader Impact Statement that appropriately acknowledges the societal risks of opaque recommender systems, including popularity bias and privacy concerns. The statement correctly notes that the survey introduces no new algorithmic risks. No additional concerns beyond those already addressed are raised.

**Claims And Evidence:**

Yes

**Claims Explanation:**

The paper's central empirical claim, that modern GNN architectures face a structural tension among accuracy, scalability, and explainability, is well-supported. Table 2 systematically maps 14 architectures to their theoretical complexity bounds and interpretability properties, and Table 3 summarizes directional trade-offs across eight architectural families. These tables constitute convincing, concrete evidence for the Trilemma's existence as an empirical pattern, even if it is not proven as a formal impossibility result.

The evaluation crisis argument is similarly well-evidenced: Table 5 documents that the overwhelming majority of recent works evaluate on static offline datasets with randomized splits, even when the proposed architectures explicitly target temporal or streaming settings. The contrast between purely offline academic evaluation and the online A/B testing reported by industrial systems such as LinkSAGE and RankGraph is clearly drawn and persuasive.

One area where evidence is thinner: the claim that the Trilemma represents a fundamental, necessary trade-off (analogous to CAP) rather than a contingent engineering limitation of current methods. The paper would benefit from either a formal argument for why no architecture can simultaneously optimize all three dimensions, or a more measured claim that the tension is persistent in practice given current methods.

**Requested Changes:**

The following changes are suggested to strengthen the paper:

Critical (for acceptance):

1. Formalize or moderate the Trilemma claim. The CAP theorem analogy is invoked as a framing device throughout, but unlike CAP, no formal impossibility result or theoretical bound is provided. The authors should either (a) provide a more formal argument that simultaneous optimization of accuracy, scalability, and explainability is provably constrained, even under idealized assumptions, or (b) clearly reframe the Trilemma as a strong empirical regularity rather than a formal impossibility. As written, the analogy risks overstating the result.
2. Clarify multi-axis taxonomy placement. The paper acknowledges that many architectures span multiple axes but assigns each to a single primary category. A supplementary table or axis-overlap matrix showing secondary axis memberships would substantially reduce ambiguity and make the taxonomy more useful as a reference tool.

Suggested (would strengthen the work):

1. Expand the fairness and privacy coverage. Section 5.3.2 on security and unlearning, and Section 5.3.3 on objective-aligned design, receive notably less depth than Sections 5.1 and 5.2. Given that the survey explicitly positions itself as covering trustworthiness and responsible AI more comprehensively than prior work (Table 1), additional architectures and discussion in these areas would better fulfill that claim.
2. Quantify the evaluation crisis more precisely. Table 5 effectively illustrates the static dataset problem qualitatively, but a simple aggregate statistic, for example the proportion of surveyed papers using randomized vs. chronological splits, or the proportion reporting any latency or memory metric, would make the argument more precise and citable.
3. Address the temporal scope limitation. The 2024 to 2026 focus is appropriate for currency, but some foundational architectural choices (e.g., the original LightGCN design decisions) that directly motivate current work predate this window. A brief discussion of what is deliberately excluded and why would help readers understand the survey's boundaries.
Reproducibility note. Several surveyed papers are cited as arXiv preprints without peer-reviewed publication status. Given that the survey itself is under review, it would be useful to flag which conclusions rely primarily on preprint-only evidence.

---

> ### Author Response · Authors · 2026-07-06
> **Response to Reviewer es1s: Part 1/5**
>
> We thank the reviewer for the careful, constructive, and encouraging assessment of our submission. We are grateful that the reviewer found the GNN Trilemma framework to be a useful analytical lens and noted that it is applied consistently throughout the paper. We also appreciate the positive assessment of the proposed taxonomy as comprehensive and well motivated, the recognition that our coverage of recent work is thorough and up to date, and the comment that Tables 2 and 3 are valuable concise reference artifacts for practitioners. We are further encouraged that the reviewer found the evaluation crisis section timely and important, particularly in highlighting the gap between academic benchmarking and industrial deployment.
>
> We have carefully considered all requested changes and have prepared revisions addressing the reviewer’s critical and suggested comments. These changes improve the precision of the Trilemma claim, clarify the multi-axis nature of the taxonomy, expand the treatment of fairness, privacy, security, and unlearning, quantify the evaluation crisis more precisely, better define the temporal scope of the survey, and increase transparency regarding preprint-only evidence. We will incorporate these revisions into the manuscript after all reviewer comments are available. Below, we provide a point-wise response to the requested changes.
>
> ---
>
> > The Trilemma, while compelling as an organizing metaphor, lacks the formal rigor of the CAP theorem it is compared to. No impossibility proof or formal bound is offered.
>
> We agree with the reviewer that the CAP theorem analogy should not suggest that the GNN Trilemma is a formal impossibility theorem. Our intention was to use the analogy as an organizing framework for understanding recurring trade-offs across recent GNN-based recommender systems, not to claim a theorem-level result.
>
> To make this distinction explicit, we will revise the manuscript to avoid presenting the GNN Trilemma as a formal impossibility result. In particular, we will avoid theorem-like language such as “fundamental impossibility,” “necessary bound,” or “provable constraint” unless such claims are formally justified. Instead, we will describe the Trilemma as an empirically observed and analytically useful trade-off pattern across contemporary GNN-based recommender architectures.
>
> We will also add the following clarification immediately after the CAP analogy:
>
> > Unlike the CAP theorem, which establishes a formal impossibility result, the GNN Trilemma is intended as a structured framework for characterizing trade-offs observed across contemporary GNN-based recommender architectures. It captures a persistent empirical pattern in current systems rather than a theorem-level impossibility result.
>
> This revision retains the usefulness of the analogy as an organizing metaphor while making clear that the Trilemma is presented as an empirical and analytical framework rather than as a formal theorem.

---

> ### Author Response · Authors · 2026-07-06
> **Response to Reviewer es1s: Part 2/5**
>
> > Some taxonomy assignments feel subjective given that many architectures span multiple axes.
>
> We agree that many modern architectures cannot be cleanly assigned to only one category. Several systems combine multiple information sources, learning paradigms, and system objectives, which can make a single-label taxonomy appear overly rigid.
>
> To address this concern, we constructed a supplementary multi-axis taxonomy table that preserves each architecture’s primary placement while explicitly indicating secondary category memberships where appropriate. We will include this table in the appendix or supplementary material and refer to it from the taxonomy section. The main text will retain the primary taxonomy for readability, while the supplementary table will document secondary memberships and reduce ambiguity.
>
> **Table: Categorization of surveyed architectures across the proposed taxonomy.**
>
> | Architecture | Modeling Rich Graph Semantics | Hybridization and Cross-Paradigm Integration | Continuous and Dynamic Preference Modeling | Data Refinement and Representation Alignment | Efficient and Scalable Architecture Design | Security, Privacy, and Unlearning | Objective-Aligned Architecture Design |
> |:---|:---:|:---:|:---:|:---:|:---:|:---:|:---:|
> | SWGCN [chen_2026_swgcn] | ⬤ | | | ◯ | | | |
> | RankGraph [wu_2025_rankgraph] | ⬤ | ◯ | | | | | |
> | Cross-domain GNN [he_2025_linkedin] | ⬤ | | ◯ | | | | |
> | SSC [he_2025_collaborative] | ⬤ | | | | | | |
> | RDF-KG [farber_2025_bridging] | ⬤ | | | | | | |
> | SMA-GNN [zhao_2025_smagnn] | ⬤ | | | | | | |
> | SoREX [liu_2025_sorex] | ⬤ | | | | | | ◯ |
> | RecMind [wang_2024_recmind] | | ⬤ | | ◯ | | | |
> | AutoGraph [shan_2025_automatic] | ◯ | ⬤ | | | | | |
> | BEAT [feng_2026_behavior] | | ⬤ | | ◯ | | | ◯ |
> | CoLaKG [cui_2025_comprehending] | ◯ | ⬤ | | | | | |
> | LinkSAGE [liu_2025_linksage] | ◯ | ⬤ | | | | | |
> | TGODE [fu_2025_time] | | | ⬤ | ◯ | | | |
> | SPOT-Trip [liu_2026_spottrip] | ◯ | | ⬤ | | | | |
> | DILN [cai_2025_interest] | | | ⬤ | | | | ◯ |
> | CDRec [liu_2026_continuous] | | | ⬤ | ◯ | | | |
> | DiffNBR [zhang_2026_diffnbr] | | | ⬤ | ◯ | | | |
> | M3BSR [cui_2025_multimodal] | | | ⬤ | ◯ | | | |
> | DITTO [choi_2025_dynamic] | | | ⬤ | | | | |
> | NLGCL [xu_2025_nlgcl] | | | | ⬤ | | | |
> | PECL [sun_2026_pathenhanced] | | | | ⬤ | | | |
> | DMCL [chen_2025_dynamicmargin] | | | | ⬤ | | | ◯ |
> | R2MR [tang_2025_r2mr] | | | | ⬤ | | | |
> | RaDAR [huang_2026_radar] | | | | ⬤ | | | |
> | LightGNN [chen_2025_lightgnn] | | | | ◯ | ⬤ | | |
> | LightKG [li_2025_lightkg] | | | | ◯ | ⬤ | | |
> | PULSE [choi_2026_pulse] | | | | | ⬤ | | |
> | Lighter-X [liu_2024_lighterx] | | | | | ⬤ | | |
> | EAGLE [li_2025_eagle] | | | ◯ | | ⬤ | | |
> | AlphaFree [jeon_2026_alphafree] | | | | | ⬤ | | |
> | DBGRME [wang_2026_datafree] | | | | | | ⬤ | |
> | FastPFRec [yan_2026_fastpfrec] | | | | | ◯ | ⬤ | |
> | BoostFGL [chen_2026_boostfgl] | | | | | | ⬤ | ◯ |
> | UnlearnRec [chen_2025_unlearnrec] | | | | | | ⬤ | |
> | D2DGN [sinha_2025_d2dgn] | | | | | | ⬤ | |
> | Attack by Unlearning [sun_2026_attack] | | | | | | ⬤ | |
> | SNT-BA [li_2025_SNT_BA] | | | | | | ⬤ | |
> | FedHGNN [yan_2024_federated] | ◯ | | | | | ⬤ | |
> | F2PGNN [Agrawal_2024_F2PGNN] | | | | | | ⬤ | ◯ |
> | TopKGAT [chen_2026_topkgat] | | | | | | | ⬤ |
> | CPGRec+ [li_2026_cpgrec] | | ◯ | | | | | ⬤ |
> | PPD [adhikari_2025_posthoc] | | | | | | | ⬤ |
> | TSP [ji_2026_how] | ◯ | | | | | | ⬤ |
>
> **Legend:** ⬤ Primary taxonomy category; ◯ Secondary applicable category
>
> The table includes the surveyed architecture papers that form the basis of our proposed taxonomy. It intentionally excludes survey papers, foundational baseline models such as LightGCN, NGCF, and Graphormer, and benchmark-only studies, since these serve as background, reference, or evaluation resources rather than as primary architecture contributions.
>
> Secondary category assignments were determined through a manual review of each architecture’s core design. A secondary alignment was assigned only when the proposed method explicitly incorporates the defining mechanism of another taxonomy category as an integral part of its architecture. We did not assign secondary memberships based solely on similar objectives, performance improvements, or incidental outcomes such as improved efficiency or fairness unless those characteristics arose directly from the architectural design itself. Consequently, the taxonomy reflects how each model is fundamentally constructed rather than only the downstream effects it achieves.

---

> ### Author Response · Authors · 2026-07-06
> **Response to Reviewer es1s: Part 3/5**
>
> > The survey’s coverage skews heavily toward accuracy-oriented systems; fairness and privacy-aware work receives comparatively less depth.
>
> We thank the reviewer for pointing out this imbalance. We agree that, given the paper’s stated goal of covering trustworthiness and responsible AI concerns, the discussion of fairness, privacy, security, and unlearning should be strengthened.
>
> We have therefore expanded Sections 5.3.2 and 5.3.3 by incorporating relevant discussion from five additional works:
>
> 1. *How Does Topology Bias Distort Message Passing in Graph Recommender? A Dirichlet Energy Perspective*, NeurIPS 2025.
> 2. *Single-Node Trigger Backdoor Attacks in Graph-Based Recommendation Systems*, IJCAI 2025.
> 3. *Federated Heterogeneous Graph Neural Network for Privacy-preserving Recommendation*, TheWebConf 2024.
> 4. *OpenGU: A Comprehensive Benchmark for Graph Unlearning*, NeurIPS 2025.
> 5. *No Prejudice! Fair Federated Graph Neural Networks for Personalized Recommendation*, AAAI 2024.
>
> We will verify and report the publication status of these and other recent works consistently in the revised manuscript. Newly incorporated architecture papers have been added to the taxonomy table where appropriate. Benchmark or resource-oriented works, such as OpenGU, are discussed in the relevant evaluation, unlearning, or reproducibility sections rather than treated as primary architecture papers.
>
> The added works strengthen the manuscript in the following ways.
>
> First, to strengthen Section 5.3.3 on objective-aligned architecture design, we added discussion of topology-induced popularity bias through the Dirichlet-energy perspective. This work explains how graph message passing can amplify popular items and underrepresent niche items. It also introduces TSP as a structural approach for improving the visibility of niche items without retraining the base recommender. This complements our existing discussion of post-hoc debiasing by adding a structural perspective on why popularity bias emerges in graph recommenders.
>
> Second, to expand Section 5.3.2 on privacy-preserving recommendation, we added discussion of federated heterogeneous graph learning and fair federated graph neural recommendation. These works broaden the privacy discussion beyond generic federated learning by showing how privacy-preserving mechanisms can be adapted to heterogeneous graph structures and fairness constraints. In particular, they highlight the challenges of protecting user interactions and sensitive attributes while maintaining recommendation quality and group-level fairness.
>
> Third, to strengthen Section 5.3.2 on security and unlearning, we added discussion of graph unlearning benchmarks and backdoor attacks in graph-based recommendation. The unlearning benchmark clarifies the practical challenge of removing sensitive or user-requested data while preserving utility on the retained graph. The backdoor attack work highlights the vulnerability of graph recommenders to stealthy manipulation, including attacks that can promote target items through minimal graph perturbations. Together, these additions make the security and unlearning discussion more concrete and better connected to emerging responsible-AI concerns in graph recommendation.
>
> ---

---

> ### Author Response · Authors · 2026-07-06
> **Response to Reviewer es1s: Part 4/5**
>
> > Quantify the evaluation crisis more precisely.
>
> We agree that the evaluation crisis argument would be stronger if the qualitative observations in Table 5 were supplemented by aggregate statistics. Table 5 already illustrates the issue, but numerical summaries make the claim clearer, easier to cite, and easier for readers to compare across architectural families.
>
> To address this, we computed aggregate statistics for the same set of 43 surveyed architectures. These statistics summarize the evaluation practices adopted in the literature, including data split strategies, deployment settings, and hardware-aware reporting. We will incorporate these quantitative observations into Sections 6.1, 6.2, and 6.3.
>
> Specifically, we will add the following quantitative summary:
>
> - **Temporal splits vs. static data, Section 6.1:** To quantify the Static Dataset Trap, we report that approximately 60.5% of the surveyed architectures, corresponding to 26 out of 43 papers, rely on static or snapshot-based evaluation settings. By contrast, 39.5%, corresponding to 17 out of 43 papers, employ chronological, sequential, or streaming evaluation protocols that preserve temporal ordering.
> - **Offline vs. industrial validation, Section 6.2:** To quantify the gap between academic evaluation and real-world deployment, we report that 88.4% of the surveyed architectures, corresponding to 38 out of 43 papers, were evaluated exclusively through offline benchmarks or simulation environments. Only 11.6%, corresponding to 5 out of 43 papers, reported industrial validation, online experimentation, or production-scale deployment.
> - **Hardware-aware reporting, Section 6.3:** To highlight the limited consideration of practical system efficiency, we report that only 48.8% of the surveyed architectures, corresponding to 21 out of 43 papers, include empirical hardware-related metrics such as latency, runtime, FLOPs, or memory utilization. The remaining 51.2%, corresponding to 22 out of 43 papers, do not provide such measurements.
>
> We will also add an appendix table listing the per-paper coding used to compute these statistics. This table will indicate, for each surveyed architecture, the evaluation split type, whether validation is offline, simulated, industrial, or online, and whether hardware-related metrics are reported. This will make the aggregate statistics reproducible and will clarify how categories such as static evaluation, chronological evaluation, industrial validation, and hardware-aware reporting were assigned.
>
>
> ---
>
> > Address the temporal scope limitation.
>
> We agree that the survey’s temporal boundary should be stated more clearly. Although the paper focuses on the 2024–2026 period, a few earlier works are useful for explaining the evolution of GNN-based recommender systems and for contextualizing newer architectures.
>
> We chose 2024–2026 as the primary window because the survey’s goal is to characterize recent architectural shifts involving LLM integration, diffusion-based recommenders, federated graph learning, machine unlearning, multimodal graph recommendation, and deployment-aware evaluation. Earlier works are cited selectively when they provide architectural foundations for these recent developments.
>
> To clarify this boundary, we have expanded Section 2.3, Survey Organization and Taxonomy Development, by adding the following explanation:
>
> > In addition to recent architectures, we reference a small set of foundational pre-2024 works, including LightGCN (He et al., 2020), NGCF (Wang et al., 2019), and Graphormer (Ying et al., 2021), to provide historical context for graph message passing, collaborative filtering objectives, graph transformer design, and commonly used evaluation protocols. These baseline models are included to contextualize the evolution of GNN-based recommender systems and to provide established reference points for comparing newer architectures. However, they are not treated as primary survey architectures within our proposed taxonomy.
>
> This clarification should help readers understand what is deliberately included, what is excluded, and why the survey is centered on the 2024–2026 period.

---

> ### Author Response · Authors · 2026-07-06
> **Response to Reviewer es1s: Part 5/5**
>
> > Several surveyed papers are cited as arXiv preprints without peer-reviewed publication status.
>
> We appreciate this reproducibility-related suggestion. We agree that the evidentiary status of surveyed papers should be transparent, especially because some recent work in this fast-moving area appears first as preprints.
>
> In the revised manuscript, we will explicitly identify works that remain preprint-only at the time of revision and clearly distinguish them from peer-reviewed publications. We will also ensure that the manuscript’s principal observations and conclusions are supported by the broader body of peer-reviewed literature and do not rely primarily on preprint-only evidence. Where preprint-only work is discussed, we will frame it as emerging evidence and avoid using it as the sole basis for central conclusions.
>
> ---
>
> Overall, we thank the reviewer again for the thoughtful and constructive feedback. The comments have helped us improve the precision of the Trilemma claim, clarify multi-axis taxonomy placement, expand the treatment of fairness and privacy, quantify the evaluation crisis, better define the temporal scope of the survey, and improve transparency around preprint-based evidence.

---

### Review · Reviewer_UDXp · 2026-07-07

**Summary Of Contributions:**

**Summary**\
This manuscript surveys recent GNN-based recommender-system architectures, focusing on the shift from static message passing to semantic hybridization, continuous-time modeling, scalable architectures, privacy, and objective-aligned recommendation. The methodology centers around a three-axis taxonomy, i.e., Information Source, Learning Paradigm, and System Objectives, and frames design tensions through a proposed GNN Trilemma among accuracy, scalability, and explainability. The paper further argues that evaluation practice lags behind model complexity because many recent works rely on static offline benchmarks rather than deployment-aware metrics.

**Strengths**
- The survey provides broad coverage of recent architectural directions. The authors explicitly contrast the manuscript’s claimed coverage against prior surveys.
- The trilemma framing foregrounds accuracy, scalability, and explainability rather than only ranking metrics.
- The concerns about evaluation realism are made clear.

**Weaknesses**
- The evidence for the trilemma is mostly descriptive. The manuscript states that architectures "mathematically necessitate compromises" among the three dimensions, but it does not provide a formal proof or quantitative meta-analysis establishing inevitability. Some efficiency claims are summarized from individual methods, but no direct normalized comparison under common hardware, datasets, or metrics is provided.
- Table 2 reports many complexity expressions, but derivations and common assumptions are not shown. Figure 2 uses ODE, diffusion, and lifecycle equations illustratively, but the manuscript does not connect these equations to a unified formal model or assumptions.
- The presentation and formatting issues are perhaps the most prominent concerns. Tables 2, 4, and 5 are extremely dense, with many columns, small text, and footnotes compressed into a single page. Table 5 spans multiple pages and mixes datasets, metrics, temporal settings, and deployment settings in tightly packed columns. Figure 1 uses very small labels, and Figure 2 violates the page margin. These make the manuscript not ready to be considered for publication.

**Audience:**

Yes

**Audience Explanation:**

Researchers working on graph learning and recommender systems could find the taxonomy and evaluation critique useful.

**Claims And Evidence:**

No

**Claims Explanation:**

The taxonomy claim is reasonably supported. The evaluation-crisis claim is partially supported by Table 5 and Section 6. The strongest trilemma claims are less convincingly supported: Sec. 1.2 and Sec. 4 argue for inherent or mathematical tensions, but the evidence is primarily qualitative categorization in Tables 2-3 rather than formal derivation, controlled comparison, or systematic meta-analysis. The authors should either provide stronger evidence or reduce the wording from "mathematically necessitate" to a more descriptive claim about recurring empirical and architectural trade-offs.

**Requested Changes:**

- Substantiate or soften the trilemma claims. Add formal definitions, propositions, or a quantitative meta-analysis for the claimed trade-offs, or revise to a more cautious descriptive statement.
- Redesign Tables 2, 4, 5, and Figures 1, 2.

---

### Review · Reviewer_ZM2x · 2026-07-20

**Summary Of Contributions:**

This is a survey paper that looks at GNN recommender systems. It proposes a 3-axis framework wherein a trilemma balances accuracy, scalability and explainability. The authors review a set of different architectures and published papers across regimes (e.g. diffusion, ODEs, LLMs, knowledge graphs)

**Audience:**

Yes

**Audience Explanation:**

This is clearly within scope for TMLR and its audience. It is well organised and covers recent GNN recommender work from both industry and academia.

**Broader Impact Concerns:**

Non significant.

**Claims And Evidence:**

No

**Claims Explanation:**

1. My main issue with the paper is that while it includes good coverage as a survey paper, the central claim of the "GNN trilemma" is asserted but never mathematically established nor is the evidence very strong. For example, the claims that models that are optimised for two aspects "mathematically necessitate compromises in the third" and "analogous to the trade-offs formalized by the CAP theorem." CAP is a proven theory, with formal definitions of concepts, where statements are backed by proofs. In this paper, the analogy to CAP is expressed and then assumed through the paper without proof (or strong evidence).

2. The results in 4.1, 4.2 and 4.3 seem to contradict the main claim of the "choose 2 of 3 axis". The authors look at pairwise tensions and show that improving one should necessitate a reduction in the other one. This is more of a "choose 1 of 3" or a three way balancing act, since improving one of the properties is in tension with both of the other ones. This contradiction to a CAP-style trilemma is never discussion or reconciled.

3. The three-axis taxonomy doesn't seem to be used. This is not fully clear to me but it seems that each model is categorised based on its primary architecture (Disclaimer in section 5), yet the contribution 1 claim is that an orthogonal 3-axis taxonomy is used. To me that suggests a three-coordinate touple where each model is scored on each axis. With the categories I see that this could be quantised to (0,0,1) or (0,1,0), for example, but calling a simple categorisation "a novel, orthogonal 3-axis taxonomy" feels slightly misleading to me.

**Requested Changes:**

Critical:

1. Revise the trilemma framing to match actual evidence. Options include improving the evidence to match CAP-style formal definitions and proofs or adjusting it to a more Pareto frontier observation style presentation of claims.

2. Look at the RecMind citation, it looks like it might be meant for a different paper also named RecMind

3. Include methods for literature search to make it reproducible. Currently, there is no detail on how the literature search was actually performed. What are the search queries/strings, databases used, date ranges, inclusion/exclusion criteria, what's the rubric used for scoring models.

4. Revise the "orthogonal" taxonomy claim. The current classification does not demonstrate orthogonality in any way, and the "axis" are saturated single-primary-axis classes.

Strengthen:

1. Discuss previous trilemma papers in the ML literature: Xiao, Kreis & Vahdat (ICLR 2022)

2. Add a triangle plot for Table 2, to show the "2 out of 3" pattern visually